



# Drainage of soft cohesive sediment with and without *Phragmites australis* as an ecological engineer

Rémon M. Saaltink[1,6*], Maria Barciela-Rial[2*], Thijs van Kessel[3], Stefan C. Dekker[1,5],
Hugo J. de Boer[1], Claire Chassagne[2], Jasper Griffioen[1,4], Martin J. Wassen[1],
Johan C. Winterwerp[2]

[1]Department of Environmental Sciences, Copernicus Institute of Sustainable Development, Utrecht University,
The Netherlands.
[2]Section of Environmental Fluid Mechanics, Delft University of Technology, The Netherlands.
[3]Deltares, P.O. Box 177, 2600 MH Delft, The Netherlands.
[4]TNO Geological Survey of the Netherlands, The Netherlands.
[5]Faculty of Management, Science and Technology, Open University, Heerlen, The Netherlands
[6]HAS University of Applied Sciences, 's-Hertogenbosch, the Netherlands

*Both authors contributed equally to this work*

*Correspondence to*: Maria Barciela-Rial (M.BarcielaRial@tudelft.nl)

**Abstract.** Conventional drainage techniques are often used to speed up consolidation of fine sediment. These techniques are relatively expensive, are invasive and often degrade the natural value of the ecosystem. This paper focusses on exploring an alternative approach that uses natural processes, rather than a technological solution, to speed up drainage of soft cohesive sediment. In a controlled column experiment, we studied how *Phragmites australis* can act as an ecological engineer that enhances drainage, thereby potentially promoting sediment consolidation. We measured the dynamics of pore water pressures at 10 cm depth intervals during a 129-day period in a column with and without plants, while the water level was fixed. Water loss via evaporation was measured using Mariotte bottles and the photosynthetic processes – including plant transpiration – were measured with a LICOR photosynthesis system. The results show that several processes initiated by *P. australis* interfere with the physical processes involved in sediment drainage and consolidation. *Phragmites australis* effectively altered the pore pressure gradient via water extraction, especially between 40 and 60 cm from the bottom of the column. In this zone, daily cycles in pore pressures were observed which could directly be linked to the diurnal cycle of stomatal gas exchange. On average, water loss via evaporation and transpiration of leaves of *P. australis* amounted to 3.9 mm day$^{-1}$, whereas evaporation of bare soil amounted on average to 0.6 mm day$^{-1}$. Moreover, the depth-averaged hydraulic conductivity increased on average by 40% in presence of *P. australis*. The results presented in this study provide information needed for predictive modelling of plants as ecological engineers to speed up soil forming processes in the construction of wetlands with soft cohesive sediment.

## 1 Introduction

Cohesive sediment is progressively being used for land reclamation (e.g., Poplar Island in the Chesapeake Bay, USA; Derrick et al., 2007). The construction of wetlands with soft sediment has become increasingly important worldwide (e.g., Mitsch et al., 1998; Saaltink et al., 2018, Barciela Rial et al., submitted). Conventional drainage is often used to speed up the consolidation process when building with soft cohesive sediments (Humphrey and Holtz, 1986). The most common methods to drain these sediments include the instalment of prefabricated vertical





strip drains, horizontal vacuum drains or sand drains (Holtz, 1987; Cognon et al., 1994; Li and Rowe, 2002). Cohesive sediment deposits consolidate primarily via self-weight consolidation (e.g., Gibson et al., 1967; Been and Sills, 1981; Winterwerp and van Kesteren, 2004). This consolidation process can be sped up by artificially draining the sediment, thereby increasing the erosion threshold, vane strength, and overall stability soon after

construction (Fagherazzi and Furbish, 2001; Chen et al., 2012). However, conventional drainage techniques are relatively expensive, are invasive and may degrade the natural value of the ecosystem (e.g., via disturbance). Our current research focusses on exploring an alternative approach which uses natural processes, rather than a technological solution, to speed up drainage of soft cohesive sediment. We refer to the concept of ecological engineering, which aims to fit environmental technology with ecosystem services (Odum and Odum, 2003). This

approach foresees the use of ecological engineers that speed up processes like sediment stability, soil formation, consolidation, and soil drainage (Jones et al., 1994).

Plants are excellent examples of ecological engineers as they directly interact with the physical and chemical components in the sediment (Angers and Caron, 1998; Ehrenfeld et al., 2005). Furthermore, plants also affect the hydrodynamic stresses (e.g., Nepf, 2012). Within the soil, plants are known to 1) increase the erosion threshold

via roots (mechanical armouring) and thereby stabilize the sediment (Waldron and Dakessian, 1982; Friend et al., 2003; Reubens et al., 2007), 2) promote soil formation by oxidizing the sediment and by altering and initiating biogeochemical processes (Visser et al., 2000; Saaltink et al., 2016), 3) compact clay particles in the vicinity of roots, which promotes consolidation (Dorioz et al., 1993), 4) change the soil hydraulic properties and soil moisture content by modifying the soil pore configuration (Angers and Caron, 1998; Kodešová et al., 2006; Gerke and

Kuchenbuch, 2007) and 5) induce an overburden by their mass. Therefore, plant roots can be considered as a cost-effective alternative to conventional vertical drainage to speed up sediment consolidation and stability.

During vegetative development, the increase in transpirational water loss is compensated by an increase in water uptake by roots, which is mainly done by increasing the root surface area (Suku et al., 2014). As roots elongate, the zone in soils where water is most actively being taken up, changes as roots are more porous near

their tips (Sanderson, 1983; Zwieniecki et al., 2003). Hence, the part of the sediment that is drained by plant roots is expected to change both horizontally and vertically over time (Gerke and Kuchenbuch, 2007). Many field studies have found that suction induced by vegetation alters pore water pressure and soil water retention (e.g., Liam et al., 1998; Smethurst et al., 2006; Leung, 2015).

Although in principle the hydraulic function of a plant root resembles that of a porous pipe (Zwieniecki et al.,

2003), little is known about the potential effect of living plant roots on the consolidation process in soft cohesive sediments, especially due to the nonlinear behavior of water distribution during vegetative development. A better understanding of how and to what extent plant roots drain cohesive sediments is essential to successfully deploy plants as eco-engineers when soft cohesive sediment is used for constructing wetlands.

In the Netherlands, a project started where soft cohesive sediment is used to construct a large wetland in Lake

Markermeer. This is a dynamic and biodiverse wetland system, while the concept of ecological engineering is used to speed up ecosystem development. To combat soil erosion and speed up ecosystem development, *Phragmites australis* (common reed) has been used as the ecological engineer to enhance the consolidation process and to improve sediment stability.

In controlled column experiments, we studied in a mechanistic way how and to what extent *P. australis* can

increase drainage in soft cohesive sediment, thereby potentially promoting consolidation. Vegetated columns were



deployed as well as a control column without vegetation. This study isolates the effects of plant roots thereby enhancing the understanding of the important plant-soil interactions in terms of consolidation by means of vertical drainage. The suitability of this species as an ecological engineer to speed up the consolidation process on newly constructed wetland can be assessed accordingly and be applied to wetland construction with soft material 5 worldwide.

## 2 Material and methods

### 2.1 Experimental set-up

Consolidation experiments were conducted in perspex (methyl methacrylate) columns (inner diameter 10 cm, height 120 cm) in the fluid mechanics laboratory of Delft University of Technology in the period November 2016 10 – August 2017. To allow the control of boundary conditions, a hollow stainless-steel pipe (outer diameter 2 cm, inner diameter 1 cm) was fixed in the middle of each perspex column (see also Barciela Rial et al., 2015). This stainless-steel pipe contained Vyon 3.2D filters in its wall allowing the water resupply of the sediment columns. These filters control the water table and prevent sediment leaking into the pipe. We refer to this pipe as a drainage pipe. To induce the constant water table at the desired level in the perspex columns, the drainage pipe was 15 connected to another column which contained Markermeer water at a fixed level of 77 cm from the bottom of the column. Water flowing from this water column to the drainage pipe (i.e., because of water loss via plant transpiration and/or evaporation) was replenished from a Mariotte bottle containing Markermeer water. Figure 1 presents a sketch of this setup.

In total, six perspex columns were used in two experimental series (Table 1). Four columns with plants were 20 harvested in experiment 1 to determine root and shoot variables, while two columns were left intact and used for pore pressure measurements. One of the two columns for pore pressure measurements was left unplanted (control column), the other column was planted with reed (vegetated column). Experiment 1 was conducted in the laboratory hall from November 2016 till February 2017 (= 118 days), with temperatures fluctuating between 15°C and 20°C, averaging at 17.3°C. The average relative humidity was 72%, fluctuating between 50% and 80%. 25 Because the pressure sensors are very sensitive to small changes in temperature, and because marginal changes in the water table were recorded, it was decided to redo the experiment in a climate room. Experiment 2 took place from March 2017 till August 2017 (t = 129 days). The environmental conditions in the climate room were kept constant at the average conditions measured in the laboratory hall. A grow light in both experimental runs was installed at the same height. Because the variation in temperature and humidity in the laboratory hall was small 30 (± 5 °C), we could link the morphological root and shoot traits measured in the first experimental run to the changes in pore pressure of the sediment in the second experimental run. Hence, the vegetation development data used in this paper was obtained from experiment 1, and the pressure sensor data was obtained from experiment 2.

The columns used for pore pressure measurements were installed with 0.5 Pa Honeywell differential pressure sensors at 0.4, 10.4, 20.4, 30.4, 40.4, 50.4, 60.4, 70.4, and 80.4 cm from the base of the column. All the sensors 35 were provided with a degassing system to avoid inaccuracy induced by air. Furthermore, a filter was installed at each connection point of the sensor to measure pore water pressure, instead of total pressure. To increase the accuracy by measuring differential pressures, all the pressure sensors were connected to a reference column filled



with a constant water level, thus measuring relative over-pressures (e.g., Fig. 1). The calibration procedure of the pressure sensors is presented in the Appendix.

The perspex columns were filled with mud from Lake Markermeer, collected by dredging (coordinates 52.54622ºN; 5.38783ºE). The sediment was thoroughly mixed before adding to the columns. The bulk density of

the suspension was about 1260 kg m$^{-3}$, the gravimetric water content was 66.7% (water mass / total mass) and the initial concentration of solids was 425 g l$^{-1}$. The specific solids density was 2580 kg m$^{-3}$. The sediment was placed in the columns and remixed. After remixing, the suspension height was 118 cm in all columns.

The sediment could settle and consolidate for 14 days, during which the sediment surface lowered to 92.5 cm in the control column and to 92.3 cm in the vegetated column (but still without vegetation). Because this 2 mm

difference between the columns is likely the result of irregularities of the bed surface at the measurement location, the consolidation rates in the two columns were considered the same, thus showing reproducibility of the consolidation experiments. Before the start of the experiment, the pore water squeezed out during self-weight consolidation was removed from above the sediment without disturbing the consolidating sediment. The removal of water from above the sediment marks the start of the experiment (time = 0 days). The mean bulk density of the

sediment at $t = 0$ was 1332 kg m$^{-3}$ for both columns.

It is to be noted that the vertical dry density and pore water pressure distributions in the initial phase of an experiment are largely affected by the experimental setup and the addition of the slurry. The drainage pipe in the middle of the column was meant to control the water table during the experiments. However, part of the slurry will dewater immediately when pouring in through the porous drainage pipe, in particular lower in the pipe, while

higher in the pipe vertical drainage prevails. As the very upper part of the drainage pipe was not porous, complex pore water circulations within the soil and the drainage pipe may have been introduced, as suggested by irregular pore water distributions. We refer to Barciela Rial (2019) for more details on this aspect of our experimental setup.

At t = 0 days, three shoots (size 2 cm) of *Phragmites australis* (common reed) were transplanted into the vegetated column and the harvest columns (Table 1). A grow light (Spectrabox Gold) with a photon flux density

of 300 μmol s m$^{-2}$ was installed at a height of 123 cm above the sediment surface. The climate room was surrounded by a white cover to maximize irradiance from the grow light. A ventilator blew constantly within the climate room to ensure air circulation.

### 2.2 Data collection

Pore pressure data from the sensors were transferred to a PC by using an analogue-digital converter and stored

every second using DasyLab. Unfortunately, some data gaps occurred due to connection problems of the sensors to the computer. From the 129 experimental days, pore pressure data were recorded for 69 days. Data gaps are evenly distributed, as shown in Supplementary Fig. S4. The quality and the resolution of the data were sufficient to capture temporal changes in pore pressure due to plant transpiration.

From the pore pressure data and the water losses, the hydraulic conductivity ($k$) in both columns can be

calculated. As the horizontal spatial scales are much smaller than the vertical scales, drainage takes place preliminary in the horizontal plane (i.e., via the drainage pipe in the center of the column). Thus, the continuity equation on cylindrical coordinates was solved accounting for radial pore water flow towards the drainage pipe (Barciela Rial, 2019):





$$k(z) = \frac{\rho g Q_0 R^2}{2\pi \Delta P(z)(R^2 - r_0^{\,2})H}\left( \ln\left(\frac{R}{r_0}\right) - \frac{1}{2} + \frac{r_0^{\,2}}{2R^2} \right) \qquad (1)$$

Here, $k$ is the hydraulic conductivity in m s$^{-1}$, $\rho$ is the density of water [kg m$^{-3}$], $g$ is the acceleration of gravity [m s$^{-2}$], $\Delta P$ (in Pa) is the pressure difference between the measured pressure at the column wall (r = R) and the pressure in the porous pipe (r = r$_0$), $Q_0$ is the measured flow in m$^3$ s$^{-1}$, $R$ is the radial coordinate, $r_0$ is the radius of the drainage pipe, and $H$ is the drainage length.

At experimental time $t = 40$, 71, 88, and 102 days, one column was harvested in experimental run 1 to measure root and shoot parameters. Above-ground biomass was cut off, after which the photosynthetic area was measured immediately. Plant tissue was air-dried at 70 °C for 48 h to determine its dry weight. The leaf per mass area (LMA) could then be calculated. Samples of 5 cm sediment were serrated from the column, after which the roots were sieved from the sediment. The root surface area and the root length in each sample were determined with SmartRoot in ImageJ (Lobet et al., 2011). The dry weight mass of the roots was determined per sample after drying, following the same procedure as the aboveground biomass.

Plant transpiration and photosynthetic activity were measured on three leaves per plant per column using the Li-Cor portable photosynthesis system (LiCor 6400) at experimental time t = 41, 61, 81, and 97 days. Conditions within the Li-Cor chamber were kept constant: the ambient $CO_2$ concentration was kept at 450 ppm, the temperature in the chamber was set to 17.3°C, the relative humidity was maintained at 60% and the light intensity in the chamber was set to 1500 PAR.

### 2.3 Data collection

Photosynthetic parameters of *P. australis* were determined with the statistical package R (Duursma, 2015) to check whether plants remained healthy and were adapted to the low-light conditions in the climate room. The results in Table 2 shows that the photosynthesis rates are realistic, with a maximum rate of the Rubisco carboxylase activity ($V_{cmax}$) varying between 115 and 39.8 µmol m$^{-2}$ s$^{-1}$ and a maximum rate of the photosynthetic electron transport ($J_{max}$) varying between 161 and 72.9 µmol m$^{-2}$ s$^{-1}$. Both variables decrease in time, which indicates a decrease in leaf effectiveness when the leaves of *P. australis* mature (i.e., photosynthesis and transpiration decreases per unit leaf area). More detailed information on photosynthetic parameters is presented in Supplementary Fig. S5, which shows net $CO_2$ assimilation rates versus light intensity.

### 3 Results

### 3.1 Plant development and water loss

Table 3 shows that leaf area and leaf biomass increased in the first months to 406 cm$^2$ and 1.48 g at day 88, after which leaves started to wilt and leaf area and leaf biomass decreased to 263 cm$^2$ and 1.00 g at the end of the experiment. The plant roots proliferated throughout the column and reached the bottom of the column at the end of the experiment (84 cm, day 129). The length, the area and the biomass of the roots increased with time. Because the plants in the harvest columns did not grow at the same speed, we corrected the root area per depth interval for leaf area as measured right before harvest. The corrected root area as calculated from the four harvests is presented in Fig. 2a, showing that the root area relative to the leaf area at each depth interval increased with time. A peak is



observed from the first harvest (40 days) in the top 5 cm (1.65 cm$^2$ per unit leaf area). This is because plants invest more in their root system than in aboveground biomass after transplantation. At 40 days, root biomass increased to 1.22 g, while leaf biomass increased to only 0.17 g (Table 3).

Evaporation led to water loss in the control column, while both evaporation and plant transpiration led to water loss in the vegetated column. Figure 3 presents evaporation rates during the experiment for the control column only. All evaporation rates fall in-between 0.3 and 0.7 mm day$^{-1}$, averaging at 0.6 mm day$^{-1}$. For the vegetated column, it was difficult to separate these two mechanisms of water loss from the sediment. Although we are aware that plants alter evaporation to a minor extent via transpiration, we used the average evaporation of 0.6 mm day$^{-1}$ from the control column to calculate evapotranspiration in the vegetated column (i.e., when measuring leaf

transpiration, we added 0.6 mm to determine evapotranspiration). The evapotranspiration data are presented in Fig. 2b. This figure shows that increasing the leaf area led to a non-linear increase in water loss via evapotranspiration. The lowest measured value of 1.4 mm day$^{-1}$ corresponds to a total leaf area of 31 cm$^2$. At a leaf area of 276 cm$^2$, the highest evaporation rate was found (7.7 mm day$^{-1}$). Evapotranspiration rates do not scale linearly with leaf area, as leaves become less effective in terms of photosynthetic capacity when maturing (Table

2). The average evapotranspiration rate of 3.9 mm day$^{-1}$ found in this study closely agrees with the average evapotranspiration value of 3.7 mm day$^{-1}$ measured in reedbeds in the Teesmouth Estuary in England during the growing season (Fermor et al., 2001). Similar rates were measured in the Biebrza wetlands in Poland, averaging between 3.0 and 3.5 in the summer months (Siedlecki et al., 2016).

**3.2 Total pore pressure gradients**

We selected three phases based on the successive stages of consolidation and drainage in the experiment as well as on the sediment height presented in Fig. 8. For the first phase, we selected data of the time steps $t = 0$ and $t = 1$ days, during which fast initial consolidation occurred. We used data of the period $t = 12$-40 days for the second phase as we lack pore pressure data from $t = 2$ until $t = 11$ days (Fig. S4). During this phase, slow consolidation occurred with little influence of plant transpiration (i.e., plant roots started to grow but did not have a big impact

on pore pressure). After 40 days, the effects of plant transpiration on total pore pressure increased. Therefore, total pore pressure data of the period $t = 41$-129 is used for the third phase.

Figure 4 shows the measured total pore water pressure for all three phases for the control column and vegetated column. The pore water pressures are rather identical for the two columns for stages 1 and 2. However, the pore pressure at the top (i.e., 2.5 cm above the water table) decreased from 1.1 kPa in phase 1 to -1.0 kPa in phase 3 in

the vegetated column, while the pore pressure at the top in the control column decreased from 2.3 kPa to 0.3 kPa. The positive pore pressures above the water table in the control column are induced by the experimental set-up, as explained above. In phase 3 of the vegetated column (Fig. 4c), the pore pressure decreases remarkably from 1.2 kPa at 50 cm to -0.9 kPa at 70 cm, peaking at -1.7 kPa at t = 74 days. This reduction in pore pressure is likely caused by water uptake by plant roots as a result of an increase in total root area increased through time (Table

3), thereby increasing water uptake from the sediment.

These results show that plants altered total pore pressure especially between 40 and 60 cm from the bottom of the column by water extraction via roots. The negative pore pressures at these depths suggest that suction of water is an important process during consolidation in presence of plants. The pore water pressure profile is clearly led



by evapotranspiration and not by self-weight consolidation since the excess pore pressure decreased at the height of the active root part indicating water transport to the roots.

### 3.3 Daily cycles in pore pressure

Water is taken up by plant roots to compensate for water loss via leaf transpiration. Plants transpire especially during photosynthesis, when stomata are open for gas exchange. Hence, it is expected that pore pressures within the sediment follow a daily cycle in the presence of plant roots. Figure 5 shows pore pressures during a 6-day period. In the control column, no difference in pore pressure is observed between day and night (Fig. 5a). However, large variation is observed in the vegetated column, especially between 40 and 50 cm height from the bottom of the column (Fig. 5b). These results suggest that during the day plants effectively lower pore water pressure at the point where the roots are extracting most of the water (50 cm from the bottom of the column). During the night, the pore water pressures increased relative to day-time. This suggests that the dominant flow of water at night occurs from the drainage pipe into the sediment to compensate for the water losses during the day. A reverse cycle is visible in the vegetated column at 70 cm, indicating that during the night, water flow from the drainage pipe decreased pressure values, likely because of a lowered water table due to plant drainage during the day. It is likely that water flow was insufficient to maintain the water table at a fixed level at short time scales because of a low hydraulic conductivity.

### 3.4 Hydraulic conductivity

The measured water fluxes for the experimental columns are presented in Fig. 6 and were used for calculating the hydraulic conductivity using Equ. (1). The theoretical evaporation rate of 0.6 mm day$^{-1}$ is defined as the daily-average of the evaporation rates presented in Fig. 3. The vegetated column reached that evaporation rate after 30 days, after which the water flux increased up to 6.2 mm day$^{-1}$ via plant transpiration. The flow in the vegetated column decreased at the end of the experiment due to maturing of the leaves of *P. australis* (Table 2).

Figure 7 shows the calculated hydraulic conductivity profiles ($k(z)$ from Equ. (1)) of the vegetated column and the control column. The water loss via leaves (transpiration) is included in the hydraulic conductivity calculations because the sum of water losses is used in Equ. 1 (i.e., evaporation and transpiration). Thus, the hydraulic conductivity of the vegetated column is based on water transport in-between the soil particles plus water transport through the plant roots. In supplementary Fig. S6, the depth-averaged hydraulic conductivity of the control column and vegetated column for the duration of the experiment are presented. In the first two days of the experiment (phase 1), the hydraulic conductivity started relatively high on average (8.8 x 10$^{-9}$ m s$^{-1}$ for the vegetated column and 5.3 x 10$^{-9}$ m s$^{-1}$ for the control column). The hydraulic conductivity rapidly decreased due to self-weight consolidation in phase 2 to 1.3 x 10$^{-11}$ m s$^{-1}$ on average in the control column and to 1.1 x 10$^{-10}$ m s$^{-1}$ on average in the vegetated column. This is in line with Fig. 8, which shows that the sediment height in both columns lowered rapidly in the first 15 days. The difference in the initial hydraulic conductivities between the control column and the vegetated column might be caused by small disturbances induced when transplanting the reed seedlings at t = 0 days. The hydraulic conductivities in both columns stabilized on average to 3.2 x 10$^{-10}$ m s$^{-1}$ in the control column and 1.3 x 10$^{-9}$ m s$^{-1}$ in the vegetated column. In phase 3, the hydraulic conductivity in the vegetated column averaged at 1.9 x 10$^{-10}$ m s$^{-1}$, while the hydraulic conductivity in the control column averaged at 1.3 x 10$^{-10}$ m s$^{-1}$. Thus, the hydraulic conductivity increased with a factor 1.4 compared to the control column due to enhanced



drainage via transpiration in the phase when plants became active. Note that at t = 0, the hydraulic conductivity computed in the vegetated and non-vegetated columns differ considerably. These differences can be explained from Equ. (1). While we measure $\Delta P(z)$, we have no information on $Q(z)$, thus using its depth-average value. However, a larger or smaller $\Delta P(z)$ would affect $Q$ locally. Together with the inherent inhomogeneities in the soil,

this leads to errors in $k(z)$. These errors reduce over time, as flow rates, and thus their absolute errors, decrease over time.

### 3.5 Sediment height

Figure 8 presents the sediment height over time. Both sediment columns had almost identical sediment heights during the experiment, ranging from 92.5 cm at the beginning of the experiment down to 85.1 cm at the end. Thus,

the sediment height is the same despite a maximum root volume increase of 16 $mm^3$ $cm^{-3}$ sediment volume in the vegetated column (Table 3). The sediment height (Fig. 8) and the increase in hydraulic conductivity in the vegetated column (Fig. 7), suggest that the volume once occupied by water is being replaced by roots. Because of continuous water supply from the drainage pipe, drainage by roots did not influence the sediment height.

### 4 Discussion

#### 4.1 Altered pore pressure gradients

The results of this study showed that *P. australis* effectively alters the pore pressure gradient in soft cohesive sediments. The shape of all pressure depth profiles (Fig. 4) is comparable with typical profiles of bare silty soils (e.g., Blight, 2003). For the vegetated column, there is a sharp drop in pore pressure between 40 and 60 cm from the base of the column. In the soil layer where plant roots extracted water, we found pressures up to four times

higher than in the control column because of vegetation induced suction. Similar impacts of plants have been found by Leung et al. (2014, 2015). They showed that the air entry value (i.e., the pressure point after which air recedes into the soil pores) increased four times in presence of Ivy trees (*Schefflera heptaphylla*) compared to bare soil, presumably because of the reduction of the size of the pores (e.g., Nimmo, 2004).

The part in the column where roots extract water did not change during the experiment: pore pressure was

reduced remarkably between 40 and 60 cm from the bottom of the column. This was unexpected as roots of *P. australis* penetrated deeper in the sediment in time (Fig. 4d) and water uptake is supposed to be largely restricted to the part near the root tip (Kramer and Boyer, 1995). The fact that pore pressure below 40 cm height was relatively unaffected even though root area increased in deeper sediment layers (Fig. 2a), suggests that the changing sediment physical properties were limiting water extraction to a sediment height of 40-60 cm from the

bottom of the column. The deep rooting depth of *P. australis* is a common trait of this species and gives it an advantage over most graminoid plants sharing wetland habitats (Moore et al., 2011). According to Zhuang et al. (2001), root hydraulic characteristics co-determine where water is taken up and this depends on the pattern by which the different parts of the root contribute to the overall water transport. These root characteristics were not measured in this experiment and it is, therefore, hard to explain why the part where water was extracted did not

shift downwards in the column through time. Because of the daily cycles present at a sediment height of 50 cm, we are confident to link the observed reduction in pore pressure at this depth to water loss by root extraction (Fig. 5b). Moreover, we measured an average water loss via evapotranspiration of 3.9 mm $day^{-1}$ in the vegetated



column, whereas water loss via evaporation amounted on average to 0.6 mm day$^{-1}$ in the control column. Although pore pressure restored during the night, the reduction in pore pressure during the day was larger than the increase during the night. This – together with the fact that root area kept on increasing in the zone of water extraction – might explain why pore pressures decreased with time. During the night, the effect of recovery of the water table

is observed at a sediment height of 70 cm in the vegetated column (Fig. 5b). At this height, water flow from the drainage pipe decreased suction values during the night because of a decreased water table due to plant drainage during the day.

Though Figure 7 is not conclusive, it is to be expected that deeper into the soil, where density increases, hydraulic conductivity decreases. While roots grow and penetrate the soil, they open drainage channels,

facilitating pore water flow along their wall (Orozco-López et al., 2018). Thus, we argue that below 60 cm this effective conductivity exceeds the soils own hydraulic conductivity. Water uptake at a depth greater than 60 cm would then originate from above.

Last, as discussed in the Introduction, plants are expected to enhance drainage, favourably affecting consolidation. The current experiments did not show any enhanced settling rates, though. This may be due to the

experimental setup chosen. However, there are a number of other arguments that need to be considered, though we have no data to quantify:

1. Plants can only root when the soil has gained a minimum strength. Consolidation rates are relatively large in the initial phase of consolidation, i.e. when plants cannot yet root.

2. The consolidation rate of soils is a function of its initial thickness and its material properties. Thus, the

20       consolidation rate likely has other time scales than root formation, which is a biological process. Thus, consolidation time scales and root formation time scales have to be compared in assessing the effectiveness of vegetation-induced consolidation and drainage,

3. The roots themselves strengthen the soil, thus also its resistance to consolidation. Hence, this armouring counteracts the additional drainage by the roots. Which of these two processes wins may be site-specific,

25       depending on vegetation type and soil properties and its initial conditions prior to consolidation.

## 4.2 Effects on hydraulic conductivity

The results of this study showed that *P. australis* increased the average hydraulic conductivity of the sediment by 40% compared to bare soil. But this induces merely circulation/ventilation, as consolidation has stopped. As discussed above, the overall hydraulic conductivity of a soil-plant complex likely consists of three parts: 1) the

inherent hydraulic conductivity of the soil itself, which is a function of the soil composition and its state of consolidation, 2) the drainage by the roots, enhancing pore water flows through the soil-plant complex, and 3) drainage channels along the roots or elsewhere in the soil in the form of root-induced cracks. Thus, the hydraulic properties of the soil and roots are closely coupled (Lobet et al., 2014). A mechanism by which plants increase the permeability in sediments involves the development of drainage channels of which the main driver is root

growth (Ghestem et al., 2011; Orozco-López et al., 2018). In our case, these macropores represent pores made by living or decaying roots of *P. australis* (i.e., root channels). Especially in cohesive sediments, these root channels are the dominant flow paths of water (Perillo et al., 1999) and can contribute to 70-100% of total macropore space in the top 8 cm of a sediment (Noguchi et al., 1997; Newman et al., 2004). However, a low fraction of macropores of total porosity already increases water flow of saturated soil (Beven and Germann, 1982). This is especially



relevant in artificial wetlands where fast initial consolidation is important. In our experiment, we found that the hydraulic conductivity increased only to a limited extent compared to bare soil, despite the increasing root area. Similar observations were reported by Vergani and Graf (2016), who observed stagnation in the increase of sediment permeability due to root proliferation when root length densities approached 0.1 cm cm$^{-3}$. This can be
explained by two opposing processes taking place when roots proliferate in the sediment: 1) the contact area of water increases with increasing root density; at low root densities this accelerates water flow through the soil, and 2) the film thickness of mobile water inside the root-induced cracks decreases with increasing root densities, deaccelerating water flow (Lange et al., 2009). Hence, a stagnant point is reached when the film thickness of the water becomes too thin to promote water flow. Another reason might be that photosynthesis and transpiration
decrease per unit leaf area as leaves mature as was observed for leaves of *P. australis* in our experiment (Table 2). The observed stagnation of the increase in hydraulic conductivity is, therefore, likely caused by a combination of a reduced photosynthetic capacity of the leaves and a reduction in film thickness.

### 4.3 Comparison with field conditions

The photosynthetic parameters measured during the experiment showed that *P. australis* behaved as expected
from field conditions; the leaves were optimized to the low-light conditions in the experimental facility. Hence, the set-up of the experiment did not affect stomatal gas exchange and data from this experiment can thus be translated to field conditions. The average evapotranspiration rate of 3.9 mm day$^{-1}$ indeed closely coincides with average evapotranspiration rates found in wetlands (e.g., Fermor et al., 2001; Siedlecki et al., 2016). Therefore, the data acquired from this experiment can be used to model the speed of drainage and consolidation in constructed
wetlands build with soft, clay-rich material. Such a model would help to estimate the difference between mudflats transplanted with and without *P. australis*. However, some complex variables were not taken into account in our experiment that will influence drainage and consolidation behavior in the field, such as variations in the topography and the depth of the water table. Moreover, if vegetation develops in patches this will also result in spatially non-uniform plant-soil interactions. Furthermore, the higher the actual evapotranspiration of the plant
species, the faster the suction recovery after a rainfall event for the same root biomass (Gaerg et al., 2015). Apart from the drainage effect, vegetation also induces biogeochemical processes (Saaltink et al., 2016), which induce pedogenic processes that accelerate the maturation or ripening of the soil (e.g. Pons and Zonneveld, 1965; Barciela Rial et al., submitted). Despite these complexities, upscaling the presented results in a predictive plant-soil model will provide useful insights for the implementation of ecological-engineers, such as *P. australis,* to speed up soil
forming processes.

### 5 Conclusions

The results presented in this study identified how ecological engineers interfere with the physical processes involved in sediment drainage and consolidation. *Phragmites australis* effectively altered the pore pressure gradient in the soft, clay-rich sediment. In our experimental set-up, this is the case for the top 40 cm of the
sediment. In this zone, daily cycles in pore pressures were observed which could directly be linked to the diurnal cycle of stomatal gas exchange. On average, water loss via evaporation and transpiration of leaves of *P. australis* amounted to 3.9 mm day$^{-1}$, whereas evaporation of bare soil amounted on average to 0.6 mm day$^{-1}$. Moreover,





the depth-averaged hydraulic conductivity increased on average by 40% in presence of *P. australis*. These findings highlight the feature of this plant to act as an eco-engineer to fasten drainage in soft cohesive sediment. This might lead to enhanced consolidation rates. However, the experiments were not fully conclusive on a number of important interactions, and further dedicated experiments and measurements are required to resolve the following

questions:

1. Roots enhance the effective drainage and hydraulic conductivity of a soil-plant complex. The inherent hydraulic conductivity of the soil itself is enhanced by root-growth induced cracks, forming macropores and drainage channels, On the other hand, root-growth disturbs the soil structure locally, which may result in densification of the soil. Further, we have indications that the roots themselves enhance drainage

within the soil by promoting pore water flow along their wall.

2. Though drainage increases, this does not necessarily imply enhanced consolidation rates. The roots also strengthen the soil by armouring, as in reinforced concrete. If the latter effect wins, consolidation rates may even be retarded, as suggested by the current experiments.

*Author contributions*. MBR and RMS, TK, SCD, HJB, JG, MJW, and JCW designed the experiments and RMS and MBR carried them out. MBR, TK, CC, and JCW developed the mathematical model to calculate the hydraulic conductivity. RMS and MBR prepared the manuscript with contributions from all co-authors.

*Competing interests*. The authors declare that they have no conflict of interest.

*Acknowledgements*. This study was supported by funding from Netherlands Organization for Scientific Research (NWO), project no. 850.13.031 and 850.13.032 and the companies Boskalis and Van Oord. This manuscript was produced with the unrestricted freedom to report all results. We would also like to express our thanks to Sander de Vree, Mohammed Jafar, Armand Middeldorp, Tom Mol, Frank Kalkman, Hans Tas and Arno Doorn for their

help, support and advice during the experiment.

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




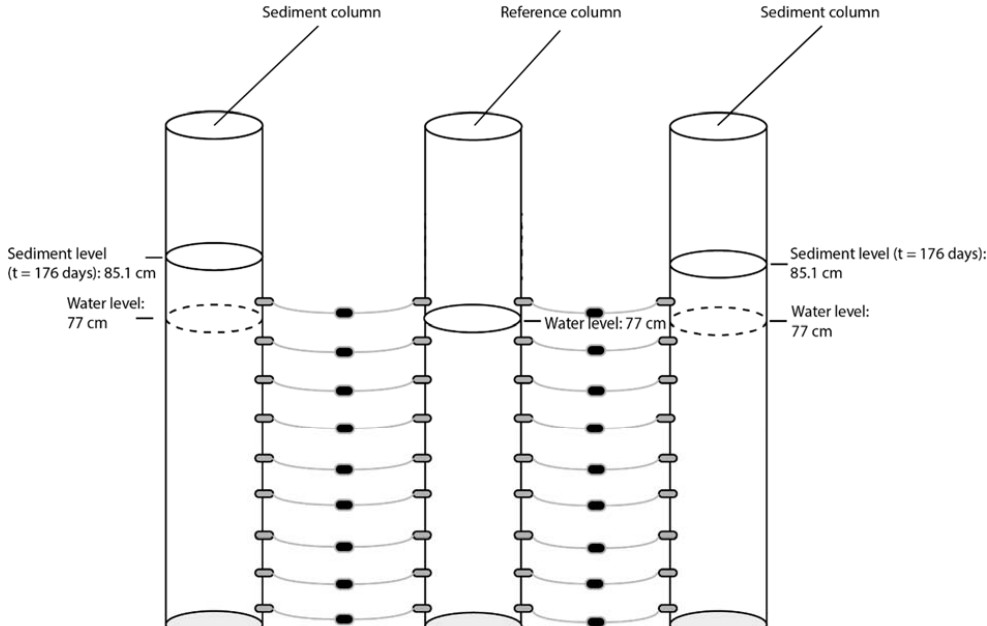

**Figure 1: Experimental set-up of the columns and location of the sensors.**





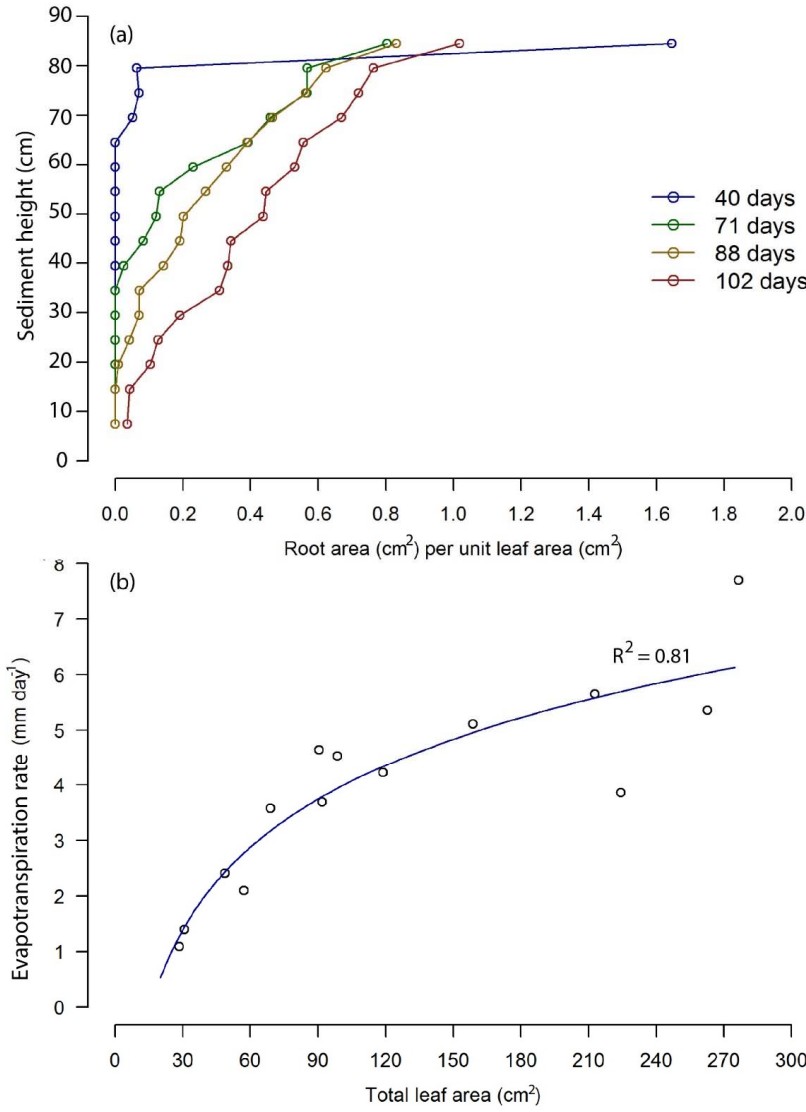

Figure 2: Root surface area per unit leaf area (cm²) across sediment height at four different time steps (a) and evapotranspiration (mm day⁻¹) as a function of total leaf area (cm²) in the vegetated column (b). Evapotranspiration rates measured in experimental run 1 and 2 are combined.




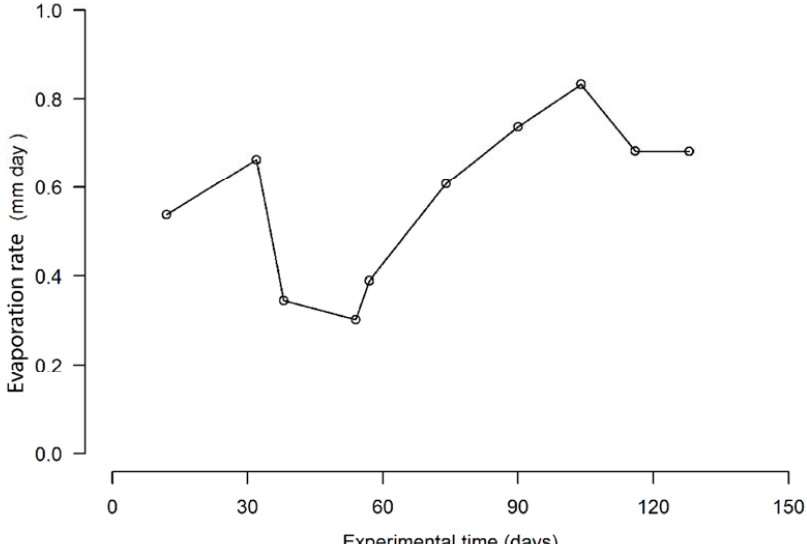

**Figure 3: Temporal change in evaporation rate for the control column (mm day⁻¹).**





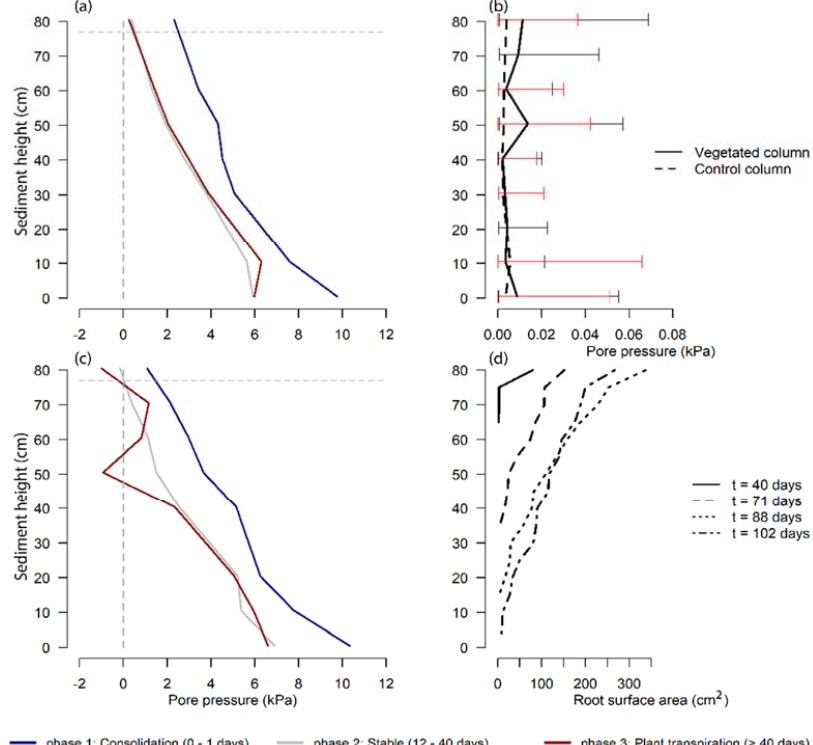

**Figure 4: Pore pressure (kPa) relative to the water column in the control column (a) and vegetated column (c). Average daily errors (kPa) for each pressure sensor are shown (± the band of measured values found during the experiment, as indicated by the bars; red is for the control column and black is for the vegetated column) (b). The average daily errors indicate the accuracy of the sensors (6.9 10⁻³ kPa) and are based on the hourly data points. Root surface area is presented from experiment 1 (d). Note that these root surface areas are from four individual plants (n = 1).**

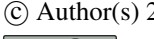


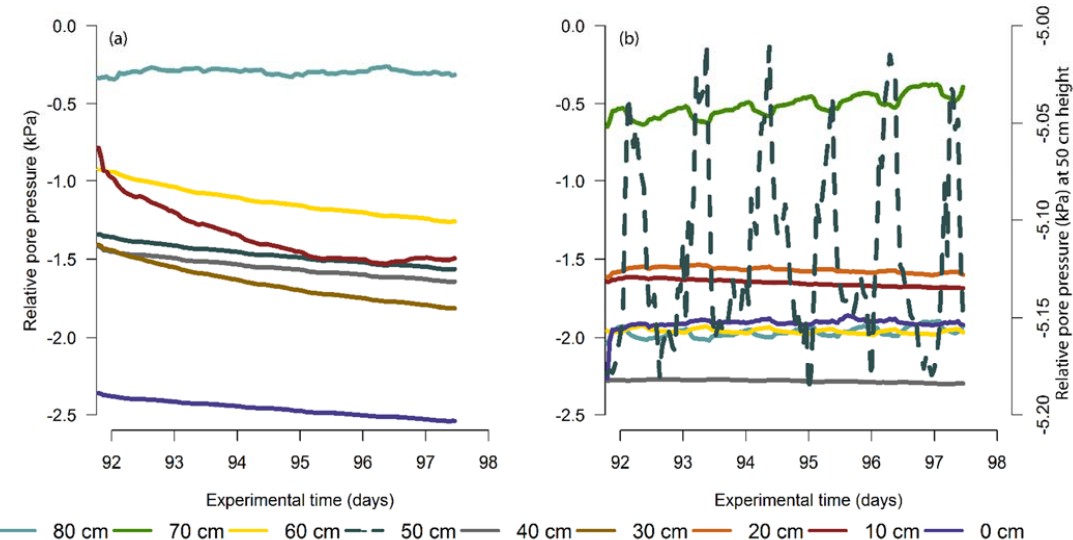

**Figure 5: Hourly time series (t = 92-98 days, to be compared with day >40 of Fig 4) of pore water pressure relative to the reference column (see Figure 1) for the control column (a) and vegetated column (b). Note that the sensor at 27 cm depth has a different y-axis in graph (b).**



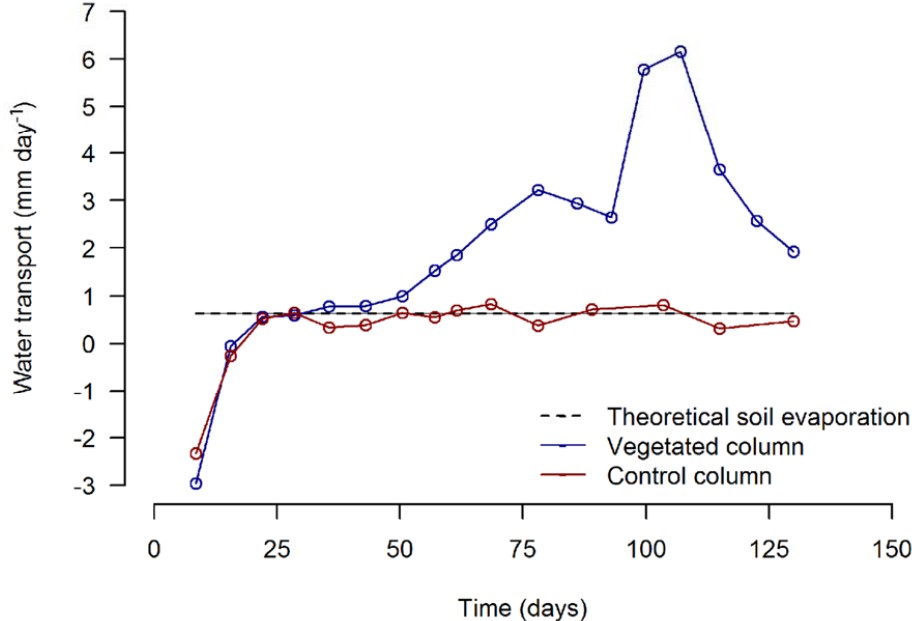

**Figure 6: Water transport measured during the experiment for the control column (consolidation or evaporation) and the vegetated column (consolidation or evaporation and plant transpiration) compared to the theoretical soil evaporation rate. Negative values indicate consolidation (dewatering via the drainage pipe) and positive values indicate evaporation (and transpiration as well).**





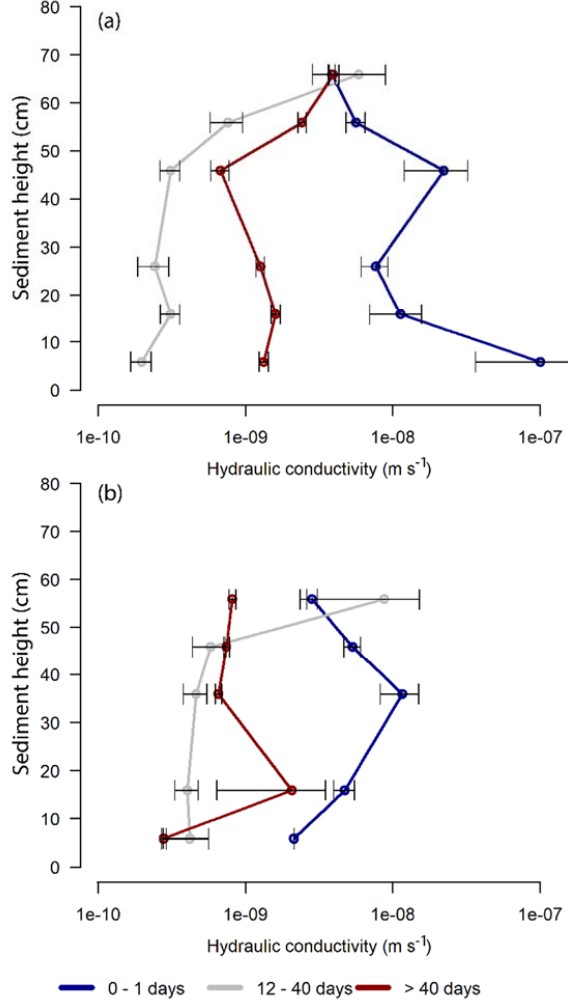

**Figure 7: Conductivity profiles (m s$^{-1}$) for the vegetated column (a) and the control column (b) with standard errors. Profiles are averaged for three different time phases: 1) fast consolidation phase (0-1 days), 2) stable phase (12-40 days), and 3) plant transpiration phase (> 40 days).**



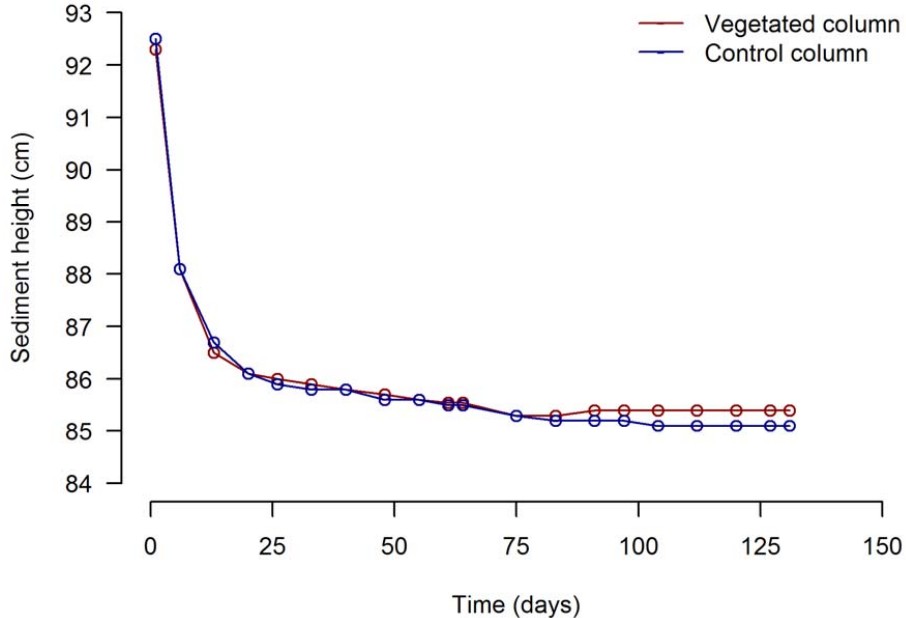

**Figure 8: Sediment height (cm) during the experiment for the vegetated column (red) and for the control column (blue).**





**Table 1: Description of the two experimental series that are part of this study.**

| Experimental series | | |
|---|---|---|
| **Experiment 1** | November 2016 – February 2017 | |
| **Condition** | *Set-up*: Laboratory hall; *Average temperature*: 17.3° C; *Relative humidity*: 50-80%; *Photon flux density (light)*: 300 μmol s m⁻²; *Water level*: fluctuating around 77 cm from the base of the column. | |
| **Column** | **Variables** | **Used in this study** |
| **1. Harvest column** | *Shoot*: leaf surface area, biomass | Yes |
| **2. Harvest column** | | Yes |
| **3. Harvest column** | *Root*: surface area, length, biomass, rooting depth | Yes |
| **4. Harvest column** | | Yes |
| **5. Vegetated column** | Pore pressure, evapotranspiration (Mariotte bottle), transpiration (Li-Cor) | No |
| **6. Control column** | Pore pressure, evaporation (Mariotte bottle). | No |
| **Experiment 2** | March – August 2017 | |
| **Condition** | *Set-up*: Climate room; *Fixed temperature*: 17.3° C; *Relative humidity*: 50-80%; *Photon flux density (light)*: 300 μmol s m⁻²; *Water level*: stabilized at 77 cm from the base of the column. | |
| **Column** | **Variables** | **Used in this study** |
| **1. Vegetated column** | Pore pressure, evapotranspiration (Mariotte bottle), transpiration (Li-Cor) | Yes |
| **2. Control column** | Pore pressure, evaporation (Mariotte bottle). | Yes |

5 **Table 2: Photosynthetic parameters of *P. australis* at 61, 81 and 97 days. The maximum rate of Rubisco carboxylase activity ($V_{cmax}$), the maximum rate of photosynthetic electron transport ($J_{max}$) and the respiration rate ($R_d$) are presented (±S.E.) as well as the light compensation point ($\Gamma^*$). All values are in μmol m⁻² s⁻¹.**

| | Day 61 | | Day 81 | | Day 97 | |
|---|---|---|---|---|---|---|
| $V_{cmax}$ | 115 | *±8.72* | 59.21 | *±3.50* | 39.8 | ±1.20 |
| $J_{max}$ | 161 | *±6.17* | 108 | *±4.84* | 72.9 | ±1.89 |
| $R_d$ | 1.67 | *±0.68* | 3.99 | *±0.55* | 0.46 | ±0.18 |
| $\Gamma^*$ | 28.79 | - | 28.84 | - | 28.63 | - |



**Table 3: Plant characteristics at 40, 71, 88 and 102 days as measured from harvested columns. Root length, root area, root biomass, and root volume are expressed per cm$^{-3}$ column volume.**

| | | 40 days | 71 days | 88 days | 102 days |
|---|---|---|---|---|---|
| Leaf area | cm$^2$ | 48.8 | 189 | 406 | 263 |
| Leaf biomass | gr | 0.17 | 0.67 | 1.48 | 1.00 |
| Leaf mass per area (LMA) | g m$^2$ | 342 | 354 | 365 | 382 |
| Stem biomass | gr | 0.43 | 1.46 | 2.13 | 2.42 |
| Max. rooting depth | cm | 18 | 48 | 68 | 80 |
| Root length | cm cm$^{-3}$ | 0.26 | 0.36 | 0.60 | 0.59 |
| Root area | cm$^2$ cm$^{-3}$ | 0.07 | 0.18 | 0.33 | 0.29 |
| Root biomass | mg cm$^{-3}$ | 0.90 | 0.80 | 1.30 | 1.07 |
| Root volume | mm$^3$ cm$^{-3}$ | 0.42 | 5.5 | 16 | 15 |
| Shoot:Root ratio | | 0.49 | 0.74 | 0.54 | 0.53 |
| Sediment volume | cm$^3$ | 6469 | 6432 | 6424 | 6414 |