# Peer review of "Drainage of soft cohesive sediment with and without *Phragmites australis* as an ecological engineer"

_Hydrology and Earth System Sciences, 2019_

## Referee Comment (RC1) · Anonymous Referee #1 · 28 Jul 2019

The manuscript presents a study to explore the use of vegetation to increase drainage with a possible effect on soil consolidation. The experimental results show that *Phragmites australis* affected pore water pressures via root water uptake, thus possibly affecting the process of drainage and consolidation.

Although the study might be interesting for the readers of HESS, I found that there are several shortcomings that, at the moment, make the manuscript unsuitable for publication in HESS.

My main concerns are:

- The presentation of the experiment and the results is not very clear. This makes it

sometime difficult to interpret results.

- It looks like parts of the results rely on data from one non-vegetated and one vegetated column (Experiment 2). Having only data from a single column in each treatment makes the result weak; a repetition of the experiment with multiple columns as done for Experiment 1 would have strengthened the results considerably.

I have listed below specific comments as they appeared in the text.

- Page 3, lines 8-18: from the description of the experiment and Figure 1 it was not very clear to me how the drainage pipe works and how it maintains a constant water table level. It is said that Figure 1 presents a sketch of the setup, but it really does not.

- P4, L22: the cited work is not in the reference list (perhaps the year is just missing in the reference list). Check all the references.

-P4, L35-38: this needs to be justified. Because it was not clear to me how the drainage pipe works, this statement was not clear as well.

- P5, subsection 2.3: the title of this subsection is the same as subsection 2.2. It is not clear why photosynthetic parameters are reported here and in Table 2. They do not seem to add any information to the study.

- P6: check the sequence of figures. Figure 3 is discussed before Figure 2b and Figure 8 is discussed here before Figure 4.

- P6, L2-3: the value of root biomass seems different from Table 1.

- Table 3: check the units of the variables. Those for the roots do not seem correct. For example, root area should be in $cm^2$.

- Figure 2a: why is root surface area per unit of leaf area reported here? The leaf area in different days is constant across the sediment height; perhaps it would be better to show just the root length or root area profile.

[Figure]

- Figure 2b: this panel seems unrelated to panel 2a. Perhaps, it would be better to have this as a panel a for Figure 3, while the current Figure 3 could be panel b for the same figure. What are the points in Fig. 3b? It would be good to differentiate between Experiment 1 and 2 (use different symbols), and explain what the points are and the time when they were estimated.

- Figures 4-8: as I understand, these figures refer to Experiment 2, with a single vegetated and a single non-vegetated column. Having only one column per treatment is not really informative, because results could just be associated with that individual experiment. I believe this is really a big limitation for the study and it makes it difficult to publish these data in a journal like HESS.

- Figure 4: the pressures shown here are relative to a reference water column. It would be good to report here the actual pore pressure instead of the relative pore pressure. As I understand, the column is completely filled with water up to 77 cm. Therefore, the pore water pressure should be positive along most of the column (apart from the top). Indicating pore pressure in the axis and having negative pressures suggests that the column became unsaturated in some parts over the vertical depth. This was rather confusing.

- Figure 5: there are more labels in the legend (every 10 cm from 0 to 80 cm) than curves. Also, the labels are different from what reported in the text and in Figure S4. Fig. 5 b reports relative pore pressures up to -5 kPa, while in Fig. 4 the pressure did not go below -2 kPa. This is not clear.

- Figure 7: panels a and b should be switch to have the same sequence (i.e., control and vegetated) as in previous figures. Also, looking at Figure S6, it is not really clear how differences in hydraulic conductivity were determined, since they appear to be roughly the same in the two treatments.

[Figure]

---

## Referee Comment (RC2) · Anonymous Referee #2 · 30 Jul 2019

General comments This paper strives to assess the use of plants for consolidation processes. I think this overall question is a relevant one, of which the results could potentially be used in nature-based restoration management. However the paper could be streamlined to improve the readability (reduce verbosity) and scientific value to the community. Generally, the graphs are not appearing successively upon mention in the text and some results (although provided) are not reported or discussed. Additionally, tenses are used inconsistently, especially in the results and discussion this needs to be improved.

Specific comments P1 line 23: It is stated that reed interferes with sediment drainage

and consolidation processes but the effects of reed on consolidation are omitted from the abstract. Either focus on purely drainage or add consolidation results.

P2 lines 29-31: This sentence indicates the knowledge gap from the previous paragraph, add it there. Lines 31-33: Integrate in the previous paragraph (same topic). Line 36: Define what you mean with ecosystem development or use sediment consolidation.

P3 Line 3: Is reed an ecosystem engineer or are you testing to see if it functions as an eco-engineer? Please be more to the point. Line 4-5: The second part of the sentence is what you want to assess but the way it is formulated seems as if this is already your conclusion. Line 13-14&17: If I understand correctly the drainage pipe is supposed to drain as well as resupply water to the accurate water table level? Please clarify. Line 16: How did you keep the water table of the water column at 77 cm, was it refilled manually? Line 17, figure 1 (P16): For a more complete overview of the experimental set-up, please include the drainage pipe and indicate if the column is vegetated. Are the black or the grey ovals your pressure sensors? How come your sediment level is measured at t=176 when your experiments ran 118 and 129 days? Line 19, table 1 (P24): if the data from column 5&6, exp 1 are not used anywhere, omit them from the table. If they are similar measurements as column 1&2, exp 2 you may be able to use them to increase your sample size for either pore pressure or sediment height. Line 37: Is your 'reference column' the same as your 'water column' ?

P4 line 3-4: Do you know anything about the benthos (such as oligochaetes and copepods) from your Markermeer mud? They may affect consolidation processes (see articles in thesis F. Cozzoli 2015). Line 16-18: Similar topic as above and below, integrate in the previous paragraph or in the section on pore pressure below data collection. Line 18-22: This seems more appropriate for the discussion. Line 33-34: Same topic, not necessary to start a new paragraph.

P5 line 17: Repeated header from p3, line 28. Line 18: statistical package Plantecophys (Duursma 2015) in R (cite R). Move to paragraph above. Line 19-25: These are results, a results section on photosynthetic parameters is missing. Move to results. Line 28-29: What were the leaf area and biomass at the start? As you planted seedlings this must have been above 0 from the start. Line 32-33: Are you referring to the 3 plants in each harvest column or between harvest columns. Generally, when you present plant data does it represent the total of the 3 plants or the average for each plant? Line 34: Figure 2 (P17) 2a and 2b seem unrelated, actually fig. 4d would be more appropriate here. Root surface area and the ratio are related graphs. Figure 2b could be a standalone figure or added to current fig 3. as you subtracted your average evaporation rate of 0.6 from all these points, correct?.

P6 line 3: Here you state 1.22g versus table 3 0.9g Line 5: figure 3 (P18) is repetitive of fig 6 but omitting the first 2 measurements. Merge information of fig 3 and 6 or just keep fig 6 and omit fig 3. Line 6: Why are you omitting the first two measurements of evaporation rate from your average calculation? Line 14: Is photosynthetic capacity a combination of V, J and R from table 2, p24? Please clarify. Line 15-18: More suitable for the discussion. Line 27: fig 4, p19 How have you calculated this total pore water pressure? The data for your 3 phases have different lengths but from the control treatment the total of phase 2&3 overlap, 28 vs 90 days of measurements. If you have multiple measurements why don't you display errorbars/uncertainty with your pore pressure data? You are often comparing vegetated and unvegetated data, maybe it would aid the reader to display control and veg in 1 graph for each phase. 4b are the daily errors similar between phases? Figure caption contains repetitive info regarding daily errors. 4d As stated above this panel may be more informative with fig 2. How can 4d contain individual plant data when 3 seedlings per column have been planted? Line 33: 50 and 70 cm data seem reversed. Line 36: If 'these results' refer to your pore pressure data then continue at previous paragraph. Line 37: Root growth mainly occurs between day 40 and 129 (fig 4b), did you check final root production data from your pore pressure vegetated column? That may differ from exp 1 and somehow explain the peak at 50 cm. Line 37-38: When looking at fig 8 the actual consolidation in that

period seems to be little, from day 40-129 less than 1 cm. How can you thus conclude that presence of reed affects consolidation?

P7 line 7: fig 5 (p20) 5a measurements from 20 and 70 cm seem to be missing. Have you chosen t= 92-98 for a specific reason? Figure caption mention phase 3 in fig 4 as comparison, 27cm should be 50cm in graph. Comparing between fig 4&5 is difficult because 4 is total and 5 is relative pore pressure.

P8 line 25-27: Did you check final root characteristics at exp 2?

P9 line 14-15: Please explain/discuss what it was in your experimental set-up that may have caused this. Line 25: provide ref for vegetation type as you only checked for reed. Provide ref for soil properties or explain how your soil properties may have affected consolidation versus other types of soils. You mentioned bulk density of the soil but for example grain size, would that affect your results? Line 27-28: In which phase did vegetation alter hydraulic conductivity or is 40% during the entire experiment.

P10 Line 23: topography of the bed? Is the water table stable in the field? How would that influence the applicability of your results in the field? Line 23-24: Why is this important? Line 28-30: The soil forming that you are discussing is mainly consolidation. From fig 8 it appears that vegetation (at least in this set-up) did not affect consolidation. The consolidation was mainly induced by physical processes in the first 15 days. Maybe this should be implemented as part of your conclusions. Line 35: specify zone (80-40 cm?).

P11 line 2: Is this (plants drain but did not affect consolidation) a side effect of the experimental set-up? Would you have expected increased consolidation with reed if you hadn't kept the water table constant? If so, discuss this at 4.3. Line 2-3: not a conclusion Line 6-13: These are not questions as stated in line 5. Line 9-10 is rep. with line 6. Do you mean to say that 1. Roots are able to enhance drainage through macropores while simultaneously reduce drainage through soil densification and an experiment should resolve what the cumulative effect is (positive or negative

for drainage). Are armouring and densification the same here? 2. Do you mean to question what else determines consolidation apart from drainage (and do you have a suggestion as to what this might be?). These 'questions' may better be posed at the end of the discussion and not in the conclusion.

Technical corrections P1 line 37: installation P2 line 25: 5) provide ref. Line 27: alters, be conclusive about direction (reduce or increase or both?) P3 line 17: Fig 1 in brackets Line 18: remove Line 26: redo should be repeat P4 line 12: pore water that sqeeuzed Line 16: remove 'It should be noted that' Line 23: please indicate the correct number of shoots for both experiments Line 24: Mention type of grow light upon first occurrence p3. Line 29: remove by Line 31: data gaps were P5 line 8: was calculated Line 17: remove Line 20: showed and were, use appropriate tense Line 28: remove 'Table 3 shows that' and add table 3 in brackets at the end of the sentence, thus focussing on the actual results. Line 31: To correspond to the graphs this should be 0 cm, day 129. And you harvested at day 102 or is this data from harvest column 5? Line 34: i.e. over time plants invest more in roots than in leaves? P6 Line 1: leaf area), because Line 6: Remove All, not correct from fig 6. Fall in > ranged Line 11: remove 'this figure shows that' and add (fig. ) at the end. Line 21: number figs successively Line 36: replace plants with reed as these results were specifically for reed. P7 line 6: replace 'in the presence of plant roots' with when vegetated Lines 7,8,10,13: is should be was / are were P9 Line 8: Although Line 13: Remove 'as discussed in the introduction' Line 14: remove 'though. This' and connect sentences Line 22: , should be . Line 24: Replace 'wins' with 'is dominant' Line 28: induced, after how many days? Line 33: Verbose, remove 'A mechanism by which' Line 35: Which macropores? This in new info reverse sentence structure P10 line 8: decelerating Line 29: remove dash Line 33: start sentence with 'In the top 40 cm,' an remove the sentence after Line 36: add (fig 5b) P11 line 35: NCK? P17 line 4: were combined. Maybe indicate exp 1&2 datapoints with color. In 2b control evaporation rates were subtracted from datapoints? P19 line 2-5: merge info on 4b. P21 line 5: remove brackets. P22 Figure 7: why do the columns differ in height? I would expect them both

to start at 80 cm, if not then at least at the same height, is the data missing? P23
Figure 8: Have you measured sediment height in your harvest columns? As the most
significant change has occurred in the first 15 days maybe you can use those data
to increase your sample size (at least for vegetated). If absolute height differed you
could use relative height to indicate the amount of consolidation (reduction from start
oid). P25 Table 3: Can you distinguish between root decay and intra plant variation?
Root length, biomass and volume decrease from 88 to 102 days while roots continue
to grow (fig 4d). This seems contrasting. Could it be the plants in harvest column 3
grew 'better' than the plants in harvest column 4? Or did something else indicate the
possibility of root decay?

Please also note the supplement to this comment:
https://www.hydrol-earth-syst-sci-discuss.net/hess-2019-194/hess-2019-194-RC2-
supplement.pdf

———————————————————————
194, 2019.

---

## Author Comment (AC1) · 7 Sep 2019

We would like to thank the two anonymous reviewers for their constructive and helpful comments on the manuscript. In the attachment, we present our point-to-point response (in red) to all comments raised (in black). We have numbered the comments made by Reviewer 1 (R 1-1 – R1-15) and Reviewer 2 (R2-1 – R2-45) for cross-referencing purposes.

References used in this response

De Lucas Pardo, M.: Effect of biota on fine sediment transport processes. A study of

lake Markermeer. PhD dissertation, Delft University, 2014.

De Lucas Pardo, M.: Tubifex worms improve densification Rates and the strengthening of soft Sediments and mine tailings. Deltares Fact-sheet EP4033.

Granéli, W.: Biomass response after nutrient addition to natural stands of reed, Phragmites australis, Internationale Vereinigung für Theoretische und Angewandte Limnologie: Verhandlungen, 22, 2956-2961, 1985.

Howard, R. J.: Intraspecific Variation in Growth of Marsh Macrophytes in Response to Salinity and Soil Type: Implications for Wetland Restoration, Estuaries and Coasts, 33, 127-138, 2009.

Jones, C. G., Lawton, J. H., Shachak, M.: Organisms as Ecosystem Engineers, Oikos, 69, 373-386, 1994.

Moore, G. E., Burdick, D. M., Peter, C. R., Keirstead, D. R.: Belowground biomass of Phragmites australis in coastal marshes, Northeastern naturalist, 19, 611-626, 2012.

Odum, E. P.: The Strategy of Ecosystem Development, Science, 164, 262-270, 1969.

Retnamony, G. R., Allam, M. M.: Effect of clay mineralogy on coefficient of consolidation, Clays and Clay minerals, 46, 596-600, 1998.

Rietkerk, M., Dekker, S.C., Wassen, M.J., Verkroost, A.W.M., Bierkens, M.F.P., 2004. A putative mechanism for bog patterning. Am. Nat. 163. doi:10.1086/383065.

Saaltink, R. M., Dekker, S. C., Griffioen, J., and Wassen, M. J.: Wetland eco-engineering: measuring and modeling feedbacks of oxidation processes between plants and clay-rich material, Biogeosciences, 13, 4945-4957, 2016.

Saaltink, R. M., Dekker, S. C., Eppinga, M. B., Griffioen, J., Wassen, M. J.: Plant-specific effects of iron toxicity in wetlands, Plant and Soil, 416, 83-96, 2017.

Saaltink, R. M., Dekker, S. C., Griffioen, J., and Wassen, M. J.: Vegetation growth

and sediment dynamics in a created freshwater wetland, Ecological Engineering, 111, 11-21, 2018.

Saaltink, R. M., Honingh, E., Dekker, S. C., Griffioen, J., Riel, M. C. van, Verdonschot, P. F. M., Vink, J. P. M., Winterwerp, J. C.; Wassen, M. J.: Respiration and aeration by bioturbating Tubificidae alter biogeochemical processes in aquatic sediment, Aquatic Sciences, 81, 2019.

Valiela, I., Teal, J. M., Persson, N. Y.: Production and dynamics of experimentally enriched salt marsh vegetation: Belowground biomass, Limnology and Oceanography, 21, 245-252, 1976.

Van Riel MC, PFM Verdonschot, DD Dekkers (2018). De bodemfauna van het Markermeer. Markermeer bodemfaunakartering 2016 en MWTL-analyse. DOI: https://doi.org/10.18174/442521

Vergani, C. and Graf, F.: Soil permeability, aggregate stability and root growth: a pot experiment from a soil bioengineering perspective, Ecohydrology, 9, 830-842, 2016.

Please also note the supplement to this comment:
https://www.hydrol-earth-syst-sci-discuss.net/hess-2019-194/hess-2019-194-AC1-supplement.pdf
* * *
[Figure]

**Supplement:**

**Response to Reviewers comments**

Saaltink & Barciela-Rial et al.

We would like to thank the two anonymous reviewers for their constructive and helpful comments on the manuscript. Here, we present our point-to-point response (in red) to all comments raised (in black). We have numbered the comments made by Reviewer 1 (R 1-1 – R1-15) and Reviewer 2 (R2-1 – R2-45) for cross-referencing purposes.

**Reviewer #1**

The manuscript presents a study to explore the use of vegetation to increase drainage with a possible effect on soil consolidation. The experimental results show that *Phragmites australis* affected pore water pressures via root water uptake, thus possibly affecting the process of drainage and consolidation. Although the study might be interesting for the readers of HESS, I found that there are several shortcomings that, at the moment, make the manuscript unsuitable for publication in HESS.

My main concerns are:

[**R1-1**]

The presentation of the experiment and the results is not very clear. This makes it sometime difficult to interpret results.

We would like to thank the reviewer for his/her specific comments below that enable us to improve the presentation and the results section of the paper. These are the following comments: [**R1-3**], [**R1-5 – R1-7**], [**R1-10**], [**R1-11**], [**R1-14**], and [**R1-15**]. Our response to these comments – as well as an explanation how we will improve the manuscript – is found below. Furthermore, Reviewer 2 also provided numerous useful suggestions to improve the presentation. For clarification, we've added a short summary at the end of this response outlining the most important changes to the Figures.

[**R1-2**]

It looks like parts of the results rely on data from one non-vegetated and one vegetated column (Experiment 2). Having only data from a single column in each treatment makes the result weak; a repetition of the experiment with multiple columns as done for Experiment 1 would have strengthened the results considerably.

We agree with the reviewer that the use of replica's in each treatment enhances the reliability of the data and allows for the quantification of variability in system responses that remain unknown in the current setup. We argue that the sensors at 9 different depths show significantly different behaviour between the vegetated and non-vegetated column during the 129 days. This total difference at every depth is called the plant effect. This plant effect can and will be different during every experiment, both in timing and in the extent at different depths. Furthermore, the plant effect can be proven without doubt owing to the closure of the water balance together with the observation of pore pressure in relation to rooting depth. Given that the plant effect is fully measured, the observed effects are replicated through time and depth (using multiple sensors).

With replicas we would be able to analyse the differences in rates of development of the root system in relation to pore pressure. In this experiment, the multiple columns with plants were used to quantify variability in plant growth (these are reported in Table 3). We are aware that the intraspecific variation in belowground biomass production can be high (e.g., Granéli, 1985; Howard 2009) and this can affect the pore pressure profile (such as presented in Figure 4). For example, the peak in Fig 4d can occur at a slightly different depth, or the pore pressures are slightly lower/higher at some depth intervals. However, the plant effect that we isolated in the experiment remains intact; the use of replica's would not have led to different conclusions and discussion points. We argue that this experiment is the first time these detailed measurements on pore pressure have been conducted, and

therefore provide a test-case for further studies that should focus on quantifying variability in the plant effect in relation to soil consolidation.

As we were aware of the risk resulting from a lack in replica's, we made sure the signals of each pressure sensor were correct. We did this by a thorough calibration of the sensors before and after the experiment. We compared these two sets of calibrations to assure that the pressure sensors were functioning properly throughout the experiment. Pressure sensors that responded differently (when comparing the second set of calibrations with the first set) were omitted from further analysis: these were the sensors installed at 10.4 cm and 60.4 cm in the control column and 50.4 cm in the vegetated column. Because of this careful calibration procedure, we are confident that the pressure profiles measured in the two treatments (and presented in the manuscript) are correct.

At last, we would like to bring forward that the non-vegetated experiments (in our case, the control column) is a classical soil mechanical experiment, carried out numerous times and published numerous times. These soil mechanical column experiments are very stable and reproducible, with only small variations in stresses and dry densities.

In the revised manuscript, we propose to add a paragraph in the discussion-section where we will describe the limitation that result from lack of replica's as well as our confidence in the plant effect that we measured.

I have listed below specific comments as they appeared in the text.
[**R1-3**]
Page 3, lines 8-18: from the description of the experiment and Figure 1 it was not very clear to me how the drainage pipe works and how it maintains a constant water table level. It is said that Figure 1 presents a sketch of the setup, but it really does not.

We agree with the reviewer that Figure 1 is lacking some important details, such as the drainage pipe. The drainage pipe is a hollow stainless-steel pipe with Vyon 3.2D filters in its wall. These filters control the water table and prevent sediment leaking into the pipe. It is important to add here that the bottom of the drainage pipe is connected to a water column that controls the water table *inside* the drainage pipe. The water in the water column is automatically replenished by a Mariotte bottle: i.e., the water table in the drainage pipe (and water column) remains constant.

We propose to 1) add the drainage pipes and the Mariotte bottles to Figure 1 and 2) expand the explanation concerning the drainage pipes in lines 8-18.

[**R1-4**]
P4, L22: the cited work is not in the reference list (perhaps the year is just missing in the reference list). Check all the references.

Here, we cited to a PhD dissertation that is currently in prep. (Barciela Rial). The dissertation will be published on 28 October 2019. We will change the year in line 22 into "in prep." in case the revised manuscript is submitted before the publication of the dissertation, or we will change in prep. in the reference list into "2019" in case the dissertation is published before the revised manuscript is submitted.

[**R1-5**]
P4, L35-38: this needs to be justified. Because it was not clear to me how the drainage pipe works, this statement was not clear as well.

See [**R1-3**]. We will extend the explanation about the drainage pipe and we will improve Figure 1. Hopefully the statement in lines 35-38 becomes clear.

[**R1-6**]
P5, subsection 2.3: the title of this subsection is the same as subsection 2.2. It is not clear why photosynthetic parameters are reported here and in Table 2. They do not seem to add any information to the study.

We thank the reviewer for pointing out this mistake. Subsection 2.3. should be called "Environmental Conditions". We will change that in the revised manuscript. We argue that the data that is presented here is important to report. As the plants were kept in laboratory conditions, we needed to make sure that these plants remained healthy and functioning properly during the experiment. This is because the driver of water flow in the column ultimately comes from photosynthesis. In the results and in the discussion section, we refer to these parameters three times: P6, line 14; P7, line 22; P10, line 10.

[**R1-7**]
P6: check the sequence of figures. Figure 3 is discussed before Figure 2b and Figure 8 is discussed here before Figure 4.

We thank the reviewer for noting this. In the revised manuscript we will thoroughly upgrade the presentation of the results. See also [R1-1] and numerous other comments made by reviewer 2. For clarification, we've added a short summary at the end of this response outlining the most important changes to the Figures.

[**R1-8**]
P6, L2-3: the value of root biomass seems different from Table 1.

This is not a mistake. The value 1.65 $cm^2$ per unit leaf area refers to the top 5 cm of the column; not the average of the column as a whole. In this line we refer to Figure 2a (not to Table 1). We will clarify this in the manuscript.

[**R1-9**]
Table 3: check the units of the variables. Those for the roots do not seem correct. For example, root area should be in $cm^2$.

The units of the variables are correct. The roots are expressed in $cm^2$ per $cm^3$ in the column to relate the root surface area per unit of soil volume. We did this on purpose as the total root area in the column is not informative. Expressing root area as $cm^2 cm^{-3}$ allowed us to compare root area with other studies, such as Vergani and Graf (2016). To avoid confusion, we will change "root area" into "root surface area per unit volume of soil" when we discuss root area in terms of $cm^2 cm^{-3}$.

[**R1-10**]
Figure 2a: why is root surface area per unit of leaf area reported here? The leaf area in different days is constant across the sediment height; perhaps it would be better to show just the root length or root area profile.

This data comes from experiment 1. At t= 40, 71, 88, and 102 days, we destroyed a column and determined the plant characteristics such as root biomass and root area. As such, the four lines in Fig 2a come from four individual plants. As biomass production is varying from individual to individual (see also [**R1-2**]), we corrected the root area with the corresponding leaf area as root:shoot ratios will be likely the same. This way of presenting the data allow us to better compare the different root areas in time.

[**R1-11**]
Figure 2b: this panel seems unrelated to panel 2a. Perhaps, it would be better to have this as a panel a for Figure 3, while the current Figure 3 could be panel b for the same figure. What are the points in Fig. 3b? It would be good to differentiate between Experiment 1 and 2 (use different symbols), and explain what the points are and the time when they were estimated.

We thank the reviewer for his suggestion to combine Fig 2b with Figure 3. The reason we did pair Fig 2a and Fig 2b is that these are data from the vegetated column, while Fig 3 presents data of the control column. This comment is also made by reviewer 2 ([**R2-22**] and [**R2-24**]). He furthermore proposes to omit Figure 3 as it essentially is presenting the same data as in Figure 6. In the revised

manuscript we will therefore omit Figure 3 and will present Figure 2a and b as a separate Figure. Figure 3 presents the evaporation rates in the control columns. These points are based on the water loss from the Mariotte bottle that was attached to the drainage pipe in the control column and were solely based on the second experiment.

**[R1-12]**
Figures 4-8: as I understand, these figures refer to Experiment 2, with a single vegetated and a single non-vegetated column. Having only one column per treatment is not really informative, because results could just be associated with that individual experiment. I believe this is really a big limitation for the study and it makes it difficult to publish these data in a journal like HESS.

This concern is also stated in comment **[R1-2]**. Our detailed response can be found there.

**[R1-13]**
Figure 4: the pressures shown here are relative to a reference water column. It would be good to report here the actual pore pressure instead of the relative pore pressure. As I understand, the column is completely filled with water up to 77 cm. Therefore, the pore water pressure should be positive along most of the column (apart from the top). Indicating pore pressure in the axis and having negative pressures suggests that the column became unsaturated in some parts over the vertical depth. This was rather confusing.

We understand the confusion of the reviewer as we put the word "relative" in the Figure caption. In fact, Figure 4 presents the actual pore pressures (Figure 5 is presenting the relative pore pressures). We agree it is confusing and we will change the caption in the next version and refer to in as "total pore water pressure".

It is true that the pore water pressure should be positive below the water table. This is indeed the case for the control column. Due to the plant effect that we isolated, the pore pressure becomes negative at 50 cm from the base of the column in the vegetated column. We will clarify this in the manuscript and clearly indicate that negative pore pressure values are caused by the water uptake via roots.

**[R1-14]**
Figure 5: there are more labels in the legend (every 10 cm from 0 to 80 cm) than curves. Also, the labels are different from what reported in the text and in Figure S4. Fig. 5 b reports relative pore pressures up to -5 kPa, while in Fig. 4 the pressure did not go below -2 kPa. This is not clear.

It is correct that there are more labels in the legend than curves. The reason is – as briefly explained in **[R1-2]** – that we omitted data for three sensors as the calibration line at the end of the experiment was different from the calibration line before the start of the experiment: at 10.4 cm and 60.4 cm in the control column and 50.4 cm in the vegetated column. All labels in the legend are used in Figure 5 a and b together, but not all labels are used in a and b separately. In the revised manuscript we will clarify the omission of the three sensors. As Figure 4 is presenting actual pore pressures and Figure 5 relative pore pressures, the values (kPa) are different in both figures. Our response at **[R1-13]** outlines how we will avoid this confusion in the revised manuscript.

**[R1-15]**
Figure 7: panels a and b should be switch to have the same sequence (i.e., control and vegetated) as in previous figures. Also, looking at Figure S6, it is not really clear how differences in hydraulic conductivity were determined, since they appear to be roughly the same in the two treatments.

We understand how this order of presenting the results can lead to confusion. We will switch panels a and b in the revised manuscript. Figure 7 presents data where the three time phases are each averaged, while Figure S6 presents the hydraulic conductivity through time. The equation we used for calculating the hydraulic conductivities can be found at P4, L34 to P5, L4.

**Reviewer #2**

**General comments**

**[R2-1]**
This paper strives to assess the use of plants for consolidation processes. I think this overall question is a relevant one, of which the results could potentially be used in nature-based restoration management. However the paper could be streamlined to improve the readability (reduce verbosity) and scientific value to the community. Generally, the graphs are not appearing successively upon mention in the text and some results (although provided) are not reported or discussed. Additionally, tenses are used inconsistently, especially in the results and discussion this needs to be improved.

We thank the reviewer for his compliment. We are also grateful for the specific and technical comments below that help us streamline the manuscript and improve the readability. Likewise, we will improve the structure and presentation (see also [**R1-1**]) and we will carefully check and improve the language.

**Specific comments**
**[R2-2]**
P1 line 23: It is stated that reed interferes with sediment drainage and consolidation processes but the effects of reed on consolidation are omitted from the abstract. Either focus on purely drainage or add consolidation results.

As the main focus of our manuscript is the effect of plants on drainage of soft cohesive sediment, we agree with the reviewer that the phrase "consolidation processes" should be omitted from the abstract. We will change line 23 in the revised manuscript.

**[R2-3]**
P2 lines 29-31: This sentence indicates the knowledge gap from the previous paragraph, add it there.

We agree. We will move the sentence "*Although in principle the hydraulic function of a plant root resembles that of a porous pipe (Zwieniecki et al., 2003), little is known about the potential effect of living plant roots on the consolidation process in soft cohesive sediments, especially due to the nonlinear behavior of water distribution during vegetative development.*" to the previous paragraph.

**[R2-4]**
Lines 31-33: Integrate in the previous paragraph (same topic).

We agree. We will make the same changes as indicated at [**R2-3**].

**[R2-5]**
Line 36: Define what you mean with ecosystem development or use sediment consolidation.

Ecosystem development can also be defined as ecological succession that results from modification of the physical environment by the organisms (Odum, 1969). When building wetlands from soft cohesive sediment, it is important to speed up this process to combat (wave and wind) erosion. We will insert the definition of Odum in the revised manuscript.

**[R2-6]**
P3 Line 3: Is reed an ecosystem engineer or are you testing to see if it functions as an ecoengineer? Please be more to the point.

An ecosystem engineer maintains and creates habitats (Jones et al., 1994). In previous studies it is confirmed that reed is functioning as an ecosystem/ecological engineer (Saaltink et al., 2016, 2017, 2018). We will clarify this in the revised manuscript.

**[R2-7]**
Line 4-5: The second part of the sentence is what you want to assess but the way it is formulated seems as if this is already your conclusion.

We agree with the reviewer that the second part is written as a conclusion. Because we also stated this in the conclusions we will remove this part from the introduction. The sentence then becomes "*The suitability of this species as an ecological engineer to speed up the consolidation process on newly constructed wetland can be assessed accordingly.*"

**[R2-8]**
Line 13-14&17: If I understand correctly the drainage pipe is supposed to drain as well as resupply water to the accurate water table level? Please clarify.

Yes, the drainage pipe can drain and resupply water. Figure 6 (copied below) makes it clear that in the beginning of the experiment, water was drained from the sediment (negative values), and after 25 days, water was resupplied via the drainage pipe. The filters in the wall of the drainage pipe control the water table and prevent sediment leaking into the pipe. It is important to add here that the bottom of the drainage pipe is connected to a water column that controls the water table *inside* the drainage pipe. The water in the water column is automatically replenished by a Mariotte bottle: i.e., the water table in the drainage pipe remains constant. As we stated at comment **[R1-3]**, we will clarify the functioning of the drainage pipe in the revised manuscript.

[Figure]

*Water transport measured during the experiment for the control column (consolidation or evaporation) and the vegetated column (consolidation or evaporation and plant transpiration) compared to the theoretical soil evaporation rate. Negative values indicate consolidation (dewatering via the drainage pipe) and positive values indicate evaporation (and transpiration as well).*

**[R2-9]**
Line 16: How did you keep the water table of the water column at 77 cm, was it refilled manually?

See our response at comment **[R1-3]** and **[R2-8]**. We will clarify this in the revised manuscript.

[**R2-10**]
Line 17, figure 1 (P16): For a more complete overview of the experimental set-up, please include the drainage pipe and indicate if the column is vegetated. Are the black or the grey ovals your pressure sensors? How come your sediment level is measured at t=176 when your experiments ran 118 and 129 days?

See our response at comment [**R1-3**]. We will add the drainage pipes and the Mariotte bottles to Figure 1. The black/grey ovals are the pressure sensors. We will make sure this is clear in the updated Figure. The duration of the experiment was indeed 118 (experiment 1) and 129 days (experiment 2). The number 176 is incorrect. We will change this in the revised Figure.

[**R2-11**]
Line 19, table 1 (P24): if the data from column 5&6, exp 1 are not used anywhere, omit them from the table. If they are similar measurements as column 1&2, exp 2 you may be able to use them to increase your sample size for either pore pressure or sediment height.

It is true that only the columns 1-4 are used from experiment 1 as the data from the pore pressure sensors were not reliable. We prefer to be very transparent about which data comes from which experiment. The Table furthermore stresses the point why it was necessary to run 2 experiments. We would very much like to keep Table 1 as is.

[**R2-12**]
Line 37: Is your 'reference column' the same as your 'water column'?

No. Although both the reference column and the water column contain Markermeer water. All the pressure sensors are connected to the reference column (the column in the middle in Fig 1.). The drainage pipes inside the sediment columns are each connected to a water column. This water column is automatically replenished by a Mariotte bottle. See also our response at comment [**R1-3**], [**R2-8-R2-10**]. The extended explanation and the improved Figure will solve this confusion.

[**R2-13**]
P4 line 3-4: Do you know anything about the benthos (such as oligochaetes and copepods) from your Markermeer mud? They may affect consolidation processes (see articles in thesis F. Cozzoli 2015).

Yes, we have knowledge about the benthos in Markermeer mud. We are aware of the study of Cozzoli and De Lucas Pardo. The soft clay-rich layer in Markermeer contains about 12,800 invertebrates $m^{-2}$, of which the Annelida take in about 40% (c. 5,000 individuals $m^{-2}$) (Van Riel et al. 2018). From these annelids, c. 3,900 individuals $m^{-2}$ belong to the subclass Oligochaeta. See also Saaltink et al. (2019). It is known that the benthos – such as oligochaetes – enhance sediment consolidation (De Lucas Pardo, 2014, 2017) up to 40-60% in the first three months. However, as the water table is below the sediment surface these processes are less relevant than evapotranspiration by plants.

[**R2-14**]
Line 16-18: Similar topic as above and below, integrate in the previous paragraph or in the section on pore pressure below data collection.

We agree with the reviewer that this will enhance the readability of the paper. We will make these changes in the revised manuscript.

[**R2-15**]
Line 18-22: This seems more appropriate for the discussion.

We agree with the reviewer. We will move this section to the discussion and merge it with the paragraph in the discussion-section where we will describe the limitation that result from lack of replica's (See comment [**R1-2**]).

[**R2-16**]
Line 33-34: Same topic, not necessary to start a new paragraph.

> We agree. We will merge these two paragraphs.

[**R2-17**]
P5 line 17: Repeated header from p3, line 28.

> See comment [R1-6]. We thank the reviewer for pointing out this mistake. Subsection 2.3. should be called "Environmental Conditions". We will change that in the revised manuscript.

[**R2-18**]
Line 18: statistical package Plantecophys (Duursma 2015) in R (cite R). Move to paragraph above.

> We agree with the reviewer that moving the sentence "*Photosynthetic parameters of P. australis were determined with the statistical package R (Duursma, 2015; R Core Team, 2018) to check whether plants remained healthy and were adapted to the low-light conditions in the climate room.*" to the end of section 2.2 is more appropriate.

[**R2-19**]
Line 19-25: These are results, a results section on photosynthetic parameters is missing. Move to results.

> We agree that moving this section to the results improves the overall structure of the paper. The section "Environmental Conditions" will be presented at the start of the Results to show that the plants remained healthy in the laboratory.

[**R2-20**]
Line 28-29: What were the leaf area and biomass at the start? As you planted seedlings this must have been above 0 from the start.

> The biomass of the plants at the start of the experiment cannot be measured without destroying the plants itself. The shoots at t = 0 did not have any leaves yet. At P4, L23-24, we explain that "At t = 0 days, three shoots (size 2 cm) of *Phragmites australis* (common reed) were transplanted into the vegetated column and the harvest columns." Hence, all shoots had the same seize at the start of the experiment.

[**R2-21**]
Line 32-33: Are you referring to the 3 plants in each harvest column or between harvest columns. Generally, when you present plant data does it represent the total of the 3 plants or the average for each plant?

> We understand the confusion made by the reviewer as we were not clear about this. We only present the total of the three plants in the column. There are three reasons for it: 1) the belowground biomass of the three individual plants cannot be separated, 2) we aimed to identify the effect on drainage and hence, total leaf area and root area per column is what matters most, and 3) as the three plants were planted very close to each other, the growth of one plant affects the other (e.g., as a result of competition for light). Therefore, it is not informative to report the average of the three plants. In the revised manuscript we will clarify this issue.

**[R2-22]**

Line 34: Figure 2 (P17) 2a and 2b seem unrelated, actually fig. 4d would be more appropriate here. Root surface area and the ratio are related graphs. Figure 2b could be a standalone figure or added to current fig 3. as you subtracted your average evaporation rate of 0.6 from all these points, correct?

The same comment was made by reviewer 1 [**R1-11**] (See also comment [**R2-24**]). In the revised manuscript we will therefore omit Figure 3 and will present Figure 2a and b as a separate Figure.

**[R2-23]**

P6 line 3: Here you state 1.22g versus table 3 0.9g

Thank you for pointing out this error. We reported the biomass with the wrong unit. It should be 0.9 mg cm$^{-3}$, instead of 1.22 g. We will change this in the revised manuscript.

**[R2-24]**

Line 5: figure 3 (P18) is repetitive of fig 6 but omitting the first 2 measurements. Merge information of fig 3 and 6 or just keep fig 6 and omit fig 3.

Figure 3 is reporting the evaporation rate from the control column. This data is used to calculate the theoretical evaporation rate for both the vegetated and the control column, which is reported in Figure 6. We agree that this may lead to confusion and we propose to omit Figure 3 in the revised manuscript as it essentially is presenting the same data as in Figure 6.

**[R2-25]**

Line 6: Why are you omitting the first two measurements of evaporation rate from your average calculation?

The evaporation rates are based on the amount of water that was released from the Mariotte bottle to the drainage pipe. In the first two measurements, water was not flowing out of the drainage pipe (resupply) but into the drainage pipe (draining the sediment column). Therefore we could not include these measurements in the calculation of the average evaporation rate.

**[R2-26]**

Line 14: Is photosynthetic capacity a combination of V, J and R from table 2, p24? Please clarify.

The reviewer is correct. Photosynthetic capacity follows from the parameters presented in Table 2: Rubisco carboxylase activity ($V_{cmax}$) and the maximum rate of photosynthetic electron transport ($J_{max}$). We will clarify this in the revised manuscript.

**[R2-27]**

Line 15-18: More suitable for the discussion.

We agree with the reviewer and will merge these lines in the discussion section.

[**R2-28**]
Line 27: fig 4, p19 How have you calculated this total pore water pressure? The data for your 3 phases have different lengths but from the control treatment the total of phase 2&3 overlap, 28 vs 90 days of measurements. If you have multiple measurements why don't you display errorbars/uncertainty with your pore pressure data? You are often comparing vegetated and unvegetated data, maybe it would aid the reader to display control and veg in 1 graph for each phase. 4b are the daily errors similar between phases? Figure caption contains repetitive info regarding daily errors. 4d As stated above this panel may be more informative with fig 2. How can 4d contain individual plant data when 3 seedlings per column have been planted?

How we calculated data coming from the pore pressures is explained in the Supplement (P1, L14-21: "*When converting the output of the pore pressure sensors in mV to kPa, we assumed a water density of 998.774 kg m$^{-3}$, corresponding to the constant lab temperature of 17.4$^o$ C. Furthermore, we worked with a gravity acceleration of 9.8125 m s$^{-1}$, corresponding to the latitude of the location of the laboratory (i.e., 52$^o$N). This means that 0.01 m of water results in a pressure increase of 0.01 m x 998.774 kg m$^{-3}$ x 9.8125 m s$^{-2}$ = 0.098004 kPa. Using the equations of the calibration lines in Fig. S2 and S3, we calculated the relative difference in pressure between the reference column and the sediment column for each pore pressure sensor: pressure (kPa) = 0.09800420 (a mV + b – 87 cm). Because the water level in the reference column was fixed at 87 cm during the experiment, changes in kPa are directly related to changes in pore pressure in the sediment columns.*"

We used three time phases to report pore pressure profiles. Since we report the average of the total pore water pressures, phase 2 and 3 may overlap in the control column. Actually, this is showing that no compelling changes occurred in the pore pressure profile in the control column, further strengthening the evidence that plants affected the pore water pressure profile in Fig 4c. We did not provide error bars in Fig4 a and c because the pore pressures are continuously changing, especially in the vegetated column. Therefore, we reported average daily errors in Fig 4b instead.

We agree with the reviewer that it aids the reader if we combine data of the vegetated column and the control column for each phase separately. We will furthermore move Fig 4d to Fig 2 as a separate panel. In the revised manuscript, Fig4a will present data of phase 1, Fig4b phase 2, Fig4c phase 3 and Fig4d will present the e daily error of the sensors. Likewise, we will update the Figure caption.

Fig4d is presenting data of the three plants in the column combined. This is because belowground biomass cannot be separated per individual (See also our response at comment [R2-21]). We will clarify this in the Figure caption (in the revised manuscript this panel moves to Fig 2).

[**R2-29**]
Line 33: 50 and 70 cm data seem reversed.

Thank you for pointing out this mistake. We will change this in the revised manuscript.

[**R2-30**]
Line 36: If 'these results' refer to your pore pressure data then continue at previous paragraph.

The reviewer is correct. We will combine these two paragraphs in the revised manuscript.

[**R2-31**]
Line 37: Root growth mainly occurs between day 40 and 129 (fig 4b), did you check final root production data from your pore pressure vegetated column? That may differ from exp 1 and somehow explain the peak at 50 cm.

Unfortunately, we were not able to determine the root biomass and area at the end of the experiment. This cannot be done without damaging the pressure sensors that were installed. For the other columns, we had to put them in the freezer overnight. We then could saw clean slices without disturbing the root system. However, the four columns that were destroyed show that roots are present

at 50 cm. While intraspecific variation in root growth exist (See comment [**R1-2**]) we are confident that plant roots are responsible for the peak observed at 50 cm in the vegetated column.

[**R2-32**]
Line 37-38: When looking at fig 8 the actual consolidation in that period seems to be little, from day 40-129 less than 1 cm. How can you thus conclude that presence of reed affects consolidation?

The settling and consolidation of fresh soft sediment can be divided in three phases: hindered settling, first phase of consolidation and second phase of consolidation (e.g., Winterwerp and Van Kesteren, 2004). During the first phase of consolidation, a soil structure is forming. Here, the effective stresses are small (while the hydraulic conductivity is large). During this phase, large settling rates (i.e., lowering of the sediment water interface as a result of consolidation) can be observed. Unfortunately, it is not yet possible to place seedlings during this initial phase. The seedlings would only float due to the low effective stress of the soil. Thus, it was necessary to start the experiment in the second phase of consolidation, where the effective stresses are large enough, with the settling rates being (and hydraulic permeability) smaller. Our manuscript focuses on the drainage effect by plants during this second phase of consolidation.

We agree that the experiments did not show any enhanced settling rates. We discussed this extensively in P9, L13-25. The lack of consolidation is likely due to the above mentioned starting point of the experiment (i.e., the second phase of consolidation). It is also important to note that no matter how much water was lost due to plant drainage, all of it was immediately replenished by the drainage pipe.

The experimental setup enabled us to isolate the drainage effect of plants, but not the effect on consolidation. However, as consolidation and drainage are related processes, we speculated briefly about its potential effects on consolidation in the discussion: the vegetation has two opposing effects. It enhances consolidation by increasing drainage and additional mass (self-weight-consolidation) and on the other hand, decreases consolidation by armouring the soil.

[**R2-33**]
P7 line 7: fig 5 (p20) 5a measurements from 20 and 70 cm seem to be missing. Have you chosen t= 92-98 for a specific reason? Figure caption mention phase 3 in fig 4 as comparison, 27cm should be 50cm in graph. Comparing between fig 4&5 is difficult because 4 is total and 5 is relative pore pressure.

See also our response to comment [**R1-2**]. As we were aware of the risk resulting from a lack in replica's, we made sure the signals of each pressure sensor were correct. We did this by a thorough calibration of the sensors before and after the experiment. We compared these two sets of calibrations to assure that the pressure sensors were functioning properly throughout the experiment. Pressure sensors that responded differently were omitted from further analysis: the sensors installed at 10.4 cm and 60.4 cm in the control column and 50.4 cm in the vegetated column. In Fig 5a, 70 cm correspond to 10.4 cm from the base of the column and 20 cm correspond to 60.4 cm from the base of the column.

In Fig5b we have chosen a random week of measurements somewhere at the end of the experiment because the day-night cycle induced by the plants become stronger in phase 3, while also shifting downwards.

Fig4 is presenting the depth of the sensors differently than Fig5. In the revised manuscript, we will change the presentation of the sensor depths in Fig 5 according to Fig4. Moreover, the reviewer is correct to note that Figure 4 is presenting total pore water pressure, while Figure 5 is presenting the relative pressures. We did this on purpose. If we would present the total pressures in Figure 5 it is not possible to visualize the temporal changes that we recorded (The range in total pressures is too large).

[**R2-34**]
P8 line 25-27: Did you check final root characteristics at exp 2?

Unfortunately this was not possible. See our explanation at comment [**R2-31**].

[**R2-35**]

P9 line 14-15: Please explain/discuss what it was in your experimental set-up that may have caused this.

We agree with the reviewer that adding an explanation is appropriate. As we explained at comment [**R2-32**], the lack of a difference on consolidation rates between the control and vegetated columns is due to the armouring effects of roots and to the experimental setup chosen, namely the continuous resupply of water in the drainage pipe. No matter how much water was lost due to plant drainage, all of it was immediately replenished by the drainage pipe. The experimental setup enabled us to isolate the drainage effect of plants, but not the effect on consolidation. However, as consolidation and drainage are related processes, we speculated briefly about its potential effects on consolidation in the discussion. We will extend this section and insert this information in the revised manuscript.

[**R2-36**]

Line 25: provide ref for vegetation type as you only checked for reed. Provide ref for soil properties or explain how your soil properties may have affected consolidation versus other types of soils. You mentioned bulk density of the soil but for example grain size, would that affect your results?

Inserting references certainly improves our statement. For *P. australis*, we will insert Moore et al. (2001). As an example for other wetland plants, we will refer to Valiela et al. (1976). For a brief discussion about the effect of different cohesive sediments on consolidation, we will refer to Retnamony and Allam (1998). See the reference list at the end of this response for the details. As the sediment was settled in the containers after dredging, we remixed the sediment and measured the bulk density of the slurry prior to the start of the experiment. The bulk density of the slurry will especially influence the pressures at the start of the experiment. For example the slurry could contain more water, which would lower the bulk density.

The effect of sediment composition and initial conditions on consolidation of Markermeer sediments is studied in Barciela-Rial (2019). There it is shown how particle size distribution and initial bulk density indeed affect the consolidation behaviour of a sediment.

[**R2-37**]

Line 27-28: In which phase did vegetation alter hydraulic conductivity or is 40% during the entire experiment.

In phase 3. We will clarify this in the revised manuscript.

[**R2-38**]

P10 Line 23: topography of the bed? Is the water table stable in the field? How would that influence the applicability of your results in the field?

No, we expect that the water table is not stable at field conditions. Likely, wetland vegetation will develop in patches, and as a consequence the groundwater level will be lower at places with vegetation (e.g., Rietkerk et al., 2004). In our experiment, the distance that water needed to travel to resupply the sediment was 5 cm (from the drainage pipe to the wall of the sensor column). In field conditions this distance can be much larger and given the low hydraulic conductivities of cohesive sediment, this may result in a local drop of the water table. However, the aim of this experiment was to understand in a mechanistic way how plants affect drainage. To isolate this effect we had to carefully control the water column. This information is important when upscaling the presented results in a predictive plant-soil model.

[**R2-39**]

Line 23-24: Why is this important?

This comment is referring to the comment above ([**R2-39**]). See our response for an explanation.

**[R2-40]**

Line 28-30: The soil forming that you are discussing is mainly consolidation. From fig 8 it appears that vegetation (at least in this set-up) did not affect consolidation. The consolidation was mainly induced by physical processes in the first 15 days. Maybe this should be implemented as part of your conclusions.

We agree with the reviewer. To avoid confusion, we must make it very clear that due to the experimental setup chosen, we could isolate how plants affect drainage, but not consolidation. See our response at comment [**R2-32**] for an explanation. We will make this clear in the revised manuscript.

**[R2-41]**

Line 35: specify zone (80-40 cm?).

We indeed refer to 80-40 cm (from the base of the column). Or, to make it clear, the top 40 cm of sediment in the column. We will include this in the revised manuscript.

**[R2-42]**

P11 line 2: Is this (plants drain but did not affect consolidation) a side effect of the experimental set-up? Would you have expected increased consolidation with reed if you hadn't kept the water table constant? If so, discuss this at 4.3.

Yes, this is due to the experimental setup chosen (See also comment [R2-32, R2-35, and R2-40]. If the water table was not kept constant, we most likely would have observed consolidation.

**[R2-43]**

Line 2-3: not a conclusion

Here, the reviewer is referring to "This might lead to enhanced consolidation rates." We agree that this is not a conclusion and will omit this sentence in the revised manuscript.

**[R2-44]**

Line 6-13: These are not questions as stated in line 5. Line 9-10 is rep. with line 6. Do you mean to say that 1. Roots are able to enhance drainage through macropores while simultaneously reduce drainage through soil densification and an experiment should resolve what the cumulative effect is (positive or negative for drainage). Are armouring and densification the same here? 2. Do you mean to question what else determines consolidation apart from drainage (and do you have a suggestion as to what this might be?). These 'questions' may better be posed at the end of the discussion and not in the conclusion.

The reviewer is correct that the last two points presented in the discussion are not questions. We will change it to "issues" instead. To avoid repetition, we suggest to remove the following sentence from the conclusion: "Roots enhance the effective drainage and hydraulic conductivity of a soil-plant complex.". Both points stated in the comment above are reflecting the two points made at the end of the conclusion. Densification and soil-armouring are two different things. The first is referring to compaction of clay/silt particles, while the latter is referring to the network of (fine) roots in the sediment. Lastly, we agree with the reviewer that these two points should be moved to the discussion. We propose do this in the revised manuscript.

**[R2-45]**

We are very grateful for the technical corrections stated below. We will implement these suggestions in the revised manuscript. As the last two points made by the reviewer are quite specific, we have written a separate response for those two points.

**Technical corrections**

- P1 line 37: installation
- P2 line 25: 5) provide ref.
- Line 27: alters, be conclusive about direction (reduce or increase or both?)
- P3 line 17: Fig 1 in brackets
- Line 18: remove
- Line 26: redo should be repeat
- P4 line 12: pore water that sqeeuzed
- Line 16: remove 'It should be noted that'
- Line 23: please indicate the correct number of shoots for both experiments
- Line 24: Mention type of grow light upon first occurrence p3.
- Line 29: remove by
- Line 31: data gaps were
- P5 line 8: was calculated
- Line 17: remove
- Line 20: showed and were, use appropriate tense
- Line 28: remove 'Table 3 shows that' and add table 3 in brackets at the end of the sentence, thus focussing on the actual results.
- Line 31: To correspond to the graphs this should be 0 cm, day 129. And you harvested at day 102 or is this data from harvest column 5?
- Line 34: i.e. over time plants invest more in roots than in leaves?
- P6 Line 1: leaf area), because
- Line 6: Remove All, not correct from fig 6. Fall in > ranged
- Line 11: remove 'this figure shows that' and add (fig. ) at the end.
- Line 21: number figs successively
- Line 36: replace plants with reed as these results were specifically for reed.
- P7 line 6: replace 'in the presence of plant roots' with when vegetated
- Lines 7,8,10,13: is should be was / are were
- P9 Line 8: Although
- Line 13: Remove 'as discussed in the introduction'
- Line 14: remove 'though. This' and connect sentences
- Line 22: , should be .
- Line 24: Replace 'wins' with 'is dominant'
- Line 28: induced, after how many days?
- Line 33: Verbose, remove 'A mechanism by which'
- Line 35: Which macropores? This in new info reverse sentence structure
- P10 line 8: decelerating
- Line 29: remove dash
- Line 33: start sentence with 'In the top 40 cm,' an remove the sentence after
- Line 36: add (fig 5b)
- P11 line 35: NCK?
- P17 line 4: were combined. Maybe indicate exp 1&2 datapoints with color. In 2b control evaporation rates were subtracted from datapoints?
- P19 line 2-5: merge info on 4b.
- P21 line 5: remove brackets.
- P22 Figure 7: why do the columns differ in height? I would expect them both to start at 80 cm, if not then at least at the same height, is the data missing?
- P23 Figure 8: Have you measured sediment height in your harvest columns? As the most

significant change has occurred in the first 15 days maybe you can use those data to increase your sample size (at least for vegetated). If absolute height differed you could use relative height to indicate the amount of consolidation (reduction from start oid).

Yes, we measured the sediment height also in our harvest columns. It is an excellent idea to use this data to increase the sample size for the vegetated column. In the revised manuscript, we will insert error bars that show the variation in Figure 8.

- P25 Table 3: Can you distinguish between root decay and intra plant variation? Root length, biomass and volume decrease from 88 to 102 days while roots continue to grow (fig 4d). This seems contrasting. Could it be the plants in harvest column 3 grew 'better' than the plants in harvest column 4? Or did something else indicate the possibility of root decay?

As the plants remained healthy during the experiment (Table 1) significant root decay likely did not cause the decline observed in Table 3. The "decline" is simply caused by intraspecific variation in biomass production. Column 3 grew better than 4. This can also be noted when looking at the leaf biomass. See our response at comment [**R1-10**] how we corrected for this variation in Figure 2 (root area per unit leaf area (cm$^2$).

**Figure summary**

As both reviewers suggested quite some changes in the presentation of the data, we've summarized all changes below for clarification.

|  | **Reviewed manuscript** | **Revised manuscript (proposed)** |
|---|---|---|
| **Figure 1** | Sketch of setup of experiment. | We will upgrade this Figure by adding drainage pipes, water columns and Mariotte bottles. |
| **Figure 2** | Root surface (panel a) and evapotranspiration (panel b). | Panel a becomes Figure 2 and panel b becomes Figure 3. |
| **Figure 3** | Evaporation rates. | This Figure will be omitted (repetition with Figure 6) |
| **Figure 4** | Pore pressure control (panel a), daily error (panel b), pore pressure vegetated (panel c), root area (panel d). | The three phases used in this study will be presented in three separate panels (a-c), including error bars. Panel d stays the same. |
| **Figure 5** | Hourly time series pore pressures. | The different heights presented in the legend will be changed in such a way that it is in line with Figure 4. |
| **Figure 6** | Water transport and evaporation. | As is. |
| **Figure 7** | Conductivity profiles vegetated (a) and control (b) | Panel a becomes panel b and vice versa. |
| **Figure 8** | Sediment height | Error bars for the vegetated column will be included. |

**References used in this response**

De Lucas Pardo, M.: Effect of biota on fine sediment transport processes. A study of lake Markermeer. PhD dissertation, Delft University, 2014.

De Lucas Pardo, M.: Tubifex worms improve densification Rates and the strengthening of soft Sediments and mine tailings. Deltares Fact-sheet EP4033.

Granéli, W.: Biomass response after nutrient addition to natural stands of reed, *Phragmites australis*, Internationale Vereinigung für Theoretische und Angewandte Limnologie: Verhandlungen, 22, 2956-2961, 1985.

Howard, R. J.: Intraspecific Variation in Growth of Marsh Macrophytes in Response to Salinity and Soil Type: Implications for Wetland Restoration, Estuaries and Coasts, 33, 127-138, 2009.

Jones, C. G., Lawton, J. H., Shachak, M.: Organisms as Ecosystem Engineers, Oikos, 69, 373-386, 1994.

Moore, G. E., Burdick, D. M., Peter, C. R., Keirstead, D. R.: Belowground biomass of *Phragmites australis* in coastal marshes, Northeastern naturalist, 19, 611-626, 2012.

Odum, E. P.: The Strategy of Ecosystem Development, Science, 164, 262-270, 1969.

Retnamony, G. R., Allam, M. M.: Effect of clay mineralogy on coefficient of consolidation, Clays and Clay minerals, 46, 596-600, 1998.

Rietkerk, M., Dekker, S.C., Wassen, M.J., Verkroost, A.W.M., Bierkens, M.F.P., 2004. A putative mechanism for bog patterning. Am. Nat. 163. doi:10.1086/383065.

Saaltink, R. M., Dekker, S. C., Griffioen, J., and Wassen, M. J.: Wetland eco-engineering: measuring and modeling feedbacks of oxidation processes between plants and clay-rich material, Biogeosciences, 13, 4945-4957, 2016.

Saaltink, R. M., Dekker, S. C., Eppinga, M. B., Griffioen, J., Wassen, M. J.: Plant-specific effects of iron toxicity in wetlands, Plant and Soil, 416, 83-96, 2017.

Saaltink, R. M., Dekker, S. C., Griffioen, J., and Wassen, M. J.: Vegetation growth and sediment dynamics in a created freshwater wetland, Ecological Engineering, 111, 11-21, 2018.

Saaltink, R. M., Honingh, E., Dekker, S. C., Griffioen, J., Riel, M. C. van, Verdonschot, P. F. M., Vink, J. P. M., Winterwerp, J. C.; Wassen, M. J.: Respiration and aeration by bioturbating Tubificidae alter biogeochemical processes in aquatic sediment, Aquatic Sciences, 81, 2019.

Valiela, I., Teal, J. M., Persson, N. Y.: Production and dynamics of experimentally enriched salt marsh vegetation: Belowground biomass, Limnology and Oceanography, 21, 245-252, 1976.

Van Riel MC, PFM Verdonschot, DD Dekkers (2018). De bodemfauna van het Markermeer. Markermeer bodemfaunakartering 2016 en MWTL-analyse. DOI: https://doi.org/10.18174/442521

Vergani, C. and Graf, F.: Soil permeability, aggregate stability and root growth: a pot experiment from a soil bioengineering perspective, Ecohydrology, 9, 830-842, 2016.